# Whole-exome sequencing of alpha-fetoprotein producing gastric carcinoma reveals genomic profile and therapeutic targets

Jun Lu[1,7], Yongfeng Ding[2,7], Yanyan Chen[1,7], Junjie Jiang[1], Yiran Chen[1], Yingying Huang[1], Mengjie Wu[1], Chengzhi Li[3], Mei Kong[3], Wenyi Zhao[4], Haohao Wang[1], Jing Zhang[1], Zhongqi Li[1], Yimin Lu[1], Xiongfei Yu[1], Ketao Jin[1], Donghui Zhou[1], Tianhua Zhou[5], Fei Teng[6], Haibin Zhang[1], Zhan Zhou [4✉], Haiyong Wang[1✉] & Lisong Teng[1✉]

Alpha-fetoprotein producing gastric carcinoma (AFPGC) is a rare and aggressive subtype of gastric cancer. However, little is known about the genomic features of this disease. We perform whole-exome sequencing analysis of AFPGC, and identify 34 significantly mutated genes. Somatic copy number alterations analysis reveals several significant focal amplifications (e.g. 19q12, 17q12) and focal deletions (e.g. 1p36.11, 9p21.3), and some of these negatively affect the patient prognosis. Comparative analyses reveal that AFPGC has distinct genomic features from gastric cancer of The Cancer Genome Atlas as well as four molecular subtypes. Several frequently altered genes with potential as therapeutic targets are identified in AFPGC. Further analysis reveals that AFPGC with amplification of *CCNE1* at 19q12 and/or *ERBB2* at 17q12 show poorer survival and more aggressive. Subsequently, based on our established patient-derived xenograft models for AFPGC, translational research is performed and the therapeutic value of targeting *CCNE1* and *ERBB2* is validated. In this work, we provide an understanding of genomic characteristics of AFPGC and propose a platform to explore and validate the genome-guided personalized treatment for this disease.

[1] Department of Surgical Oncology, The First Affiliated Hospital, School of Medicine, Zhejiang University, Hangzhou, China. [2] Department of Medical Oncology, The First Affiliated Hospital, School of Medicine, Zhejiang University, Hangzhou, China. [3] Department of Pathology, The First Affiliated Hospital, School of Medicine, Zhejiang University, Hangzhou, China. [4] Innovation Institute for Artificial Intelligence in Medicine and Zhejiang Provincial Key Laboratory of Anti-Cancer Drug Research, College of Pharmaceutical Sciences and Alibaba-Zhejiang University Joint Research Center of Future Digital Healthcare, Zhejiang University, Hangzhou, China. [5] Institute of Gastroenterology, Cancer center, Zhejiang University, Hangzhou, China. [6] Hangzhou Oncocare Co. Ltd, Hangzhou, China. [7] These authors contributed equally: Jun Lu, Yongfeng Ding, Yanyan Chen. ✉email: zhanzhou@zju.edu.cn; lanceter1@zju.edu.cn; lsteng@zju.edu.cn

α-Fetoprotein (AFP)-producing gastric carcinoma (AFPGC) is a rare and special subtype of gastric cancer (GC) associated with high liver metastasis rate and extremely poor prognosis[1,2]. According to the World Health Organization (WHO) classification of digestive system tumors (5th edition, 2019)[3], AFPGC is characterized as a rare type of GC with elevated serum AFP levels and positive immunohistochemistry (IHC) staining for AFP, accounting for 1.3–5.4% of GC[4–7].

To date, omics analyses of GC are performed to identify molecular characteristics of GC. The Cancer Genome Atlas (TCGA) investigators have published comprehensive genomic and transcriptomic analyses of GC and identified four subtypes: Epstein–Barr virus-infected (EBV), microsatellite instability (MSI), genomically stable (GS), and chromosomally unstable (chromosomal instability, CIN) tumor types[8]. Subsequently, the investigators of the Asian Cancer Research Group (ACRG) categorized GC into another four subtypes with differential prognosis[9]. Recently, some studies have focused on characterizing the molecular signatures of several special subtypes of GC, such as diffuse-type GC[10], early-onset GC[11], and EBV-positive GC[12]. These clinicopathological subtypes and their associated molecular characterization provided a foundation for better patient stratification and choice for targeted therapy. AFPGC is associated with more aggressive behavior and poorer prognosis as compared to common GC; however, there is no specialized treatment till date and lacked large-scale genomic studies. Thus, it is urgent to elucidate the molecular features and facilitate the development of specialized therapies for AFPGC.

Here we present the molecular landscape of AFPGC by performing whole-exome sequencing (WES), fluorescence in situ hybridization (FISH), and IHC in a cohort of 105 AFPGC patients till now. These results suggest aggressive behavior and distinct genomic features of AFPGC when compared to common GC. We also identify several frequently altered genes that are potentially targetable and finally evaluate these in the corresponding patient-derived xenograft (PDX) models. Our study elucidates the properties of AFPGC and provide a rationale for the development of specialized treatment.

## Result

**Patients and tumor samples**. A total of 5261 GC cases were diagnosed in our institution and then a cohort of 105 GC patients who were IHC positive for AFP were enrolled in our study (Supplementary Fig. 1). Of these, 58 paired fresh-frozen and/or formalin-fixed paraffin-embedded (FFPE) tumor tissues and adjacent normal tissues that met our criteria have undergone WES. All 105 FFPE tumor tissues underwent FISH and IHC tests. Among the 105 AFPGC patients, 79 (75.2%) were males and the median age at diagnosis was 64 years (range: 30–83 years) (Supplementary Data 1 and Supplementary Table 1). The majority (68.6%) of the patients were diagnosed at an advanced stage (Stage I/II, 33 patients and stage III/IV, 72 patients). Liver metastasis was found in 42 (40%) patients in our study. Compared to stage-matched non-AFPGC patients in our institute, AFPGC patients were found to have poorer prognoses (hazard ratio (HR) = 1.47, $P = 0.015$) (Supplementary Fig. 2a). The association between serum AFP level and prognosis of AFPGC was further investigated, and 270 ng/mL was chosen as a cutoff value according to the receiver operating characteristic curve. The results revealed that patients with high serum AFP level (>270 ng/mL) have poorer overall survival (OS) ($P = 0.017$) (Supplementary Fig. 2b, c).

**Somatic mutations and signatures of AFPGC**. WES analysis was performed on the genomic DNA of 58 GC tumors and matched normal tissues at a mean coverage of 161- and 105-fold, respectively (Supplementary Data 2). A total of 1041 somatic single-nucleotide variants (SNVs) and 300 somatic indels were identified, including 1179 synonymous and 161 nonsynonymous mutations that are corresponding to 4.08 nonsynonymous mutations per megabase of the targeted DNA (Supplementary Data 3). Thirty-four significantly mutated genes were identified in AFPGC using MuSic analysis[13] (Fig. 1a and Supplementary Table 2). Among these, the most frequently altered genes in AFPGC were TP53 (69%), PCLO (21%), CSMD3 (19%), and KMT2C (19%). Other frequently mutated genes (>9%) were also detected, such as LRP1B, SYNE1, FPR1, PRDM1, and PRKRIR. Notably, frequently mutated driver genes (such as PIK3CA, KRAS, APC) in TCGA-GC were rarely mutated in AFPGC (Fig. 1b and Supplementary Data 4). The frequency of TP53 mutation in AFPGC was significantly higher than that in TCGA-GC ($P < 0.01$). Furthermore, when comparing to TCGA subtypes, TP53 mutation rate of AFPGC was significantly higher than all subtypes other than TCGA-CIN (Fig. 1c and Supplementary Data 4). Detailed analysis showed that 64.3% of TP53 mutations detected in AFPGC occurred in the DNA-binding domain, including recurrent missense mutations R273H/C, R272M, and R282W (Fig. 1d). In addition, c.994-1 G > A (X331_splice), a splice site mutation in the oligomerization domain, was present in three cases of AFPGC but absent in TCGA-CIN or TCGA-GC (Supplementary Fig. 3). On the contrary, TP53 R175H/G, the most frequent mutation in TCGA-CIN, was not observed in AFPGC. Besides, the mutation rates of other significantly mutated genes, such as KMT2C, MDC1, FPR1, EPHA1, and SMAD4, were different between AFPGC and TCGA-CIN (Fig. 1c and Supplementary Data 4).

The predominant somatic mutation types were C:G>T:A transitions and C:G>A:T transversions (Supplementary Fig. 4). A total of three independent mutation signatures were screened out (Fig. 1e and Supplementary Data 5). Among these, Signature 1 showed association with a clock-like mutational process and was correlated with age[14]. Signature 29 was found in the cancer samples obtained from individuals with a tobacco chewing habit. Signature 17 was the hallmark signature of esophageal cancer and GC[15]. Importantly, the recurrent pattern of Signature 6 (associated with defective DNA mismatch repair) was found to be absent, which was consistent with the result that no MSI case was found in AFPGC (Supplementary Data 6).

Moreover, the contributions of these mutation signatures in AFPGC across various clinicopathological characteristics were shown in Fig. 1f and no significant relevance was observed (Supplementary Fig. 5).

**Somatic copy number alterations of AFPGC**. In total, 58 paired AFPGC tumor and normal samples were analyzed for somatic copy number alterations (SCNAs) using GISTIC 2.0. SCNAs were recurrent in AFPGC and also included amplifications containing CCNE1 (19q12), ERBB2 (17q12), CCND1 (11q13.3), MUC4 (3q29), MCL1 (1q21.3), and MYC (8q24.21), and deletion of ARID1A (1p36.11), CDKN2A (9p21.3), CREBBP (16p13.3), and SMAD4 (18q21.2) (Fig. 2a and Supplementary Data 7). Furthermore, the impact of SCNA on survival outcomes was explored in AFPGC using Kaplan–Meier survival analysis. The results indicated that patients with amplification in 19q12 or 17q12 might have a significantly worse OS rate than those without amplification (19q12, $P = 0.008$; 17q12, $P = 0.038$, Supplementary Fig. 6). In addition, patients with 8q24.21 amplification tended to show poorer prognosis ($P = 0.112$).

The comparison of significant SCNAs between AFPGC and TCGA-GC was further performed (Supplementary Fig. 7). A total

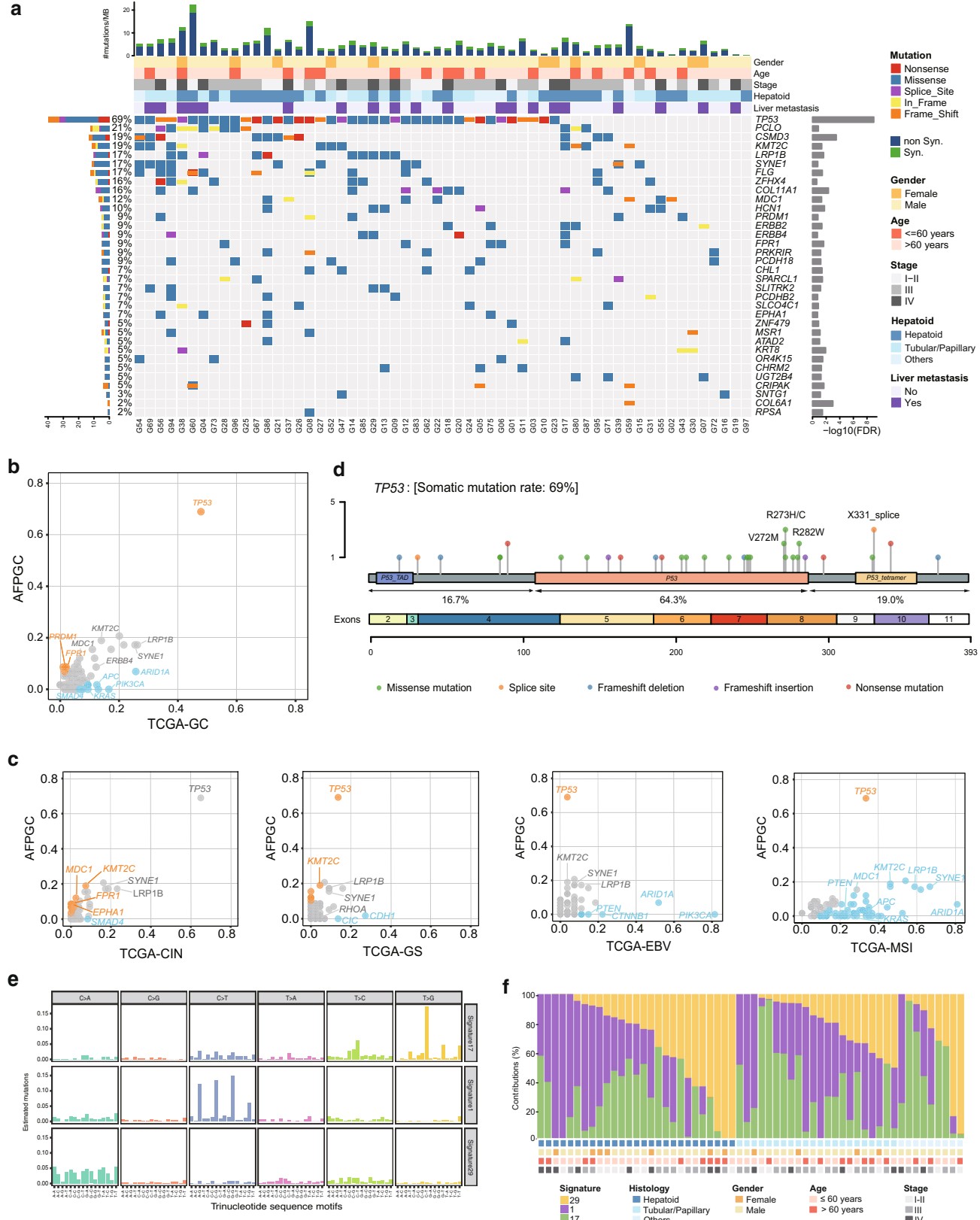

of 92 significant SCNA regions (42 amplifications and 50 deletions) were identified in AFPGC, of which 58 regions (22 amplifications and 36 deletions) were not significant in TCGA-GC, suggesting a relative specificity for AFPGC (Supplementary Fig. 8a, b). Considering that AFPGC harbored relatively higher number of TP53 mutation rate and SCNAs, we subsequently

compared the SCNAs of AFPGC with TCGA-CIN (a known subtype with recurrent SCNAs in TCGA-GC), finding that 57% (24/42) significant amplifications and 74% (37/50) deletions only existed in AFPGC (Supplementary Fig. 8c–f), such as amplifications in 7q11.21, 6p21.33, and 19q13.43, and deletions in 9q34.3 and 1p36.32 (Supplementary Fig. 7). These regions contain

**Fig. 1 The landscape of somatic mutations and mutational signatures of AFPGC. a** Somatic mutations of 58 pairs of AFPGC samples. The matrix in the middle panel shows somatic mutations by tumor sample (column) and by gene (row). The top histogram shows the frequency of nonsynonymous mutation and synonymous mutation. The top tracks show histopathological characteristics of tumor samples. The left histogram shows the number of alterations accumulated on 34 significantly mutated genes identified by MuSic analysis. The right histogram shows negative log transformation of *P*-value. **b**, **c** Gene mutation rates of AFPGC in comparison with TCGA-GC (**b**) or four subtypes of TCGA-GC (**c**). Orange dots, genes with significantly higher mutation rate in AFPGC; blue dots, genes with significantly lower mutation rate in AFPGC. **d** Distribution of nonsynonymous somatic TP53 mutations identified in 58 AFPGC. **e** Ninety-six substitutions derived from WES data of 58 pairs of AFPGC samples. The horizontal axis represents mutation patterns of 96 substitutions with different colors. The vertical axis depicts the estimated mutations attributed to a specific mutation type. **f** The distribution of mutation signatures in AFPGC across various clinicopathological characteristics. AFPGC, α-fetoprotein-producing gastric carcinoma; CIN, chromosomal instability; EBV, Epstein–Barr virus; FDR, false discovery rate; GS, genomically stable; Hepatoid, hepatoid differentiation; MSI, microsatellite instability; Non syn, nonsynonymous mutation; Syn, synonymous mutation; TCGA, The Cancer Genome Atlas; Tubular/Papillary, tubular or papillary differentiation. Source data are provided as a Source Data file.

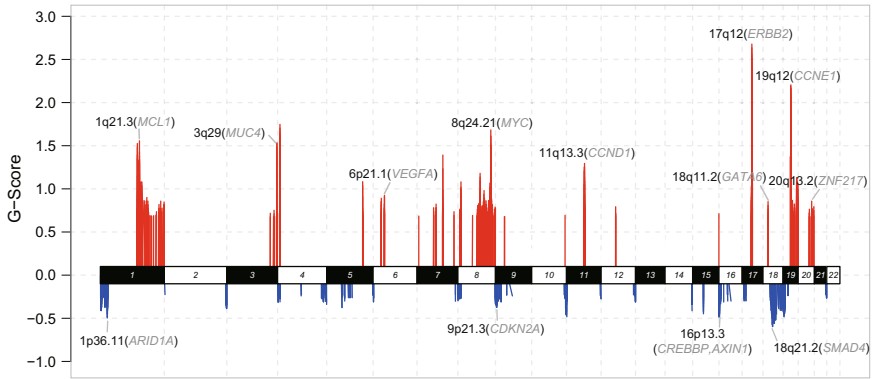

**Fig. 2 The landscape of somatic copy number alterations of AFPGC.** Somatic copy number alterations identified by GISTIC 2.0 analysis. The genome is oriented horizontally and GISTIC scores at each locus are plotted from the bottom to top. Red represents focal amplification and blue represents deletion in the genomic regions. Representative amplification/deletion and corresponding genes are labeled as the SCNA region (representative gene). G-score, GISTIC scores. Source data are provided as a Source Data file.

important cancer-related genes including *TRIM28*, *SEMA3C*, *NOTCH1*, and *FAT1*.

**Pathway analysis.** Integrated Kyoto Encyclopedia of Genes and Genomes (KEGG) pathway analysis of somatic mutations and SCNA data revealed core signaling pathway alterations in AFPGC, including RTK/RAS/PI(3)K, p53/cell cycle, and JAK/STAT (Fig. 3 and Supplementary Data 8). The RTK/RAS/PI(3)K signaling pathway was primarily altered by amplification of *ERBB2*, *ERBB3*, and *MYC*. P53/cell cycle signaling pathway was altered mainly owing to frequent mutations of *TP53* and amplification of *CCNE1*/*CCND1*, which encodes cyclins. Downstream signaling cascade genes in JAK/STAT signaling pathway, with *MCL1* as the frequently amplified gene, leads to an anti-apoptotic process. Compared to GC in the TCGA cohort (TCGA-GC), both AFPGC and TCGA-GC showed enrichment in the alterations of RTK/RAS/PI(3)K and p53/cell cycle signaling pathways, but in different patterns. In the RTK/RAS/PI(3)K signaling pathway, AFPGC showed recurrent *ERBB2* amplifications and mutations. The key components of the RAS/PI(3)K signaling pathway—*KRAS*, *PIK3CA*, and *PTEN*—showed frequent mutations in TCGA-GC, but were absent in AFPGC. In the p53/cell cycle pathway, cell cycle mediators (such as *CCNE1*, *CCND1*, and *CDK4/6*) were frequently activated in AFPGC, whereas to a significantly lesser degree in TCGA-GC. More *TP53* mutations were harbored in AFPGC, but there were fewer mutations of upstream mediators (*ATM/ATR/EP300*) and DNA repair regulators (*BRCA1/2*) as compared to TCGA-GC.

**Comparative analysis and survival analysis of potentially targetable genes.** According to the genomic alterations in AFPGC,

there were a number of alterations in known cancer-related or potentially targetable genes, including *ERBB2* (31%), *CCNE1* (29%), *MYC* (22%), *CCND1* (16%), *MCL1* (14%), *FLT1*(12%), *ERBB3* (10%), *AURKA* (10%), *AXL* (9%), *BCL6* (9%), *BRCA2* (9%), *EGFR* (9%), *ERBB4* (9%), *FGFR2* (9%), and so on, and these were comparable to a certain extent with that of the TCGA database (Fig. 4a). Representative clinical trials or preclinical studies of potential therapies targeting these genes were listed in Supplementary Table 3. Of these, *ERBB2* and *CCNE1* were found to have the highest alteration rates (31% and 29%) in AFPGC and were significantly higher than those (19% and 13%) present in the TCGA cohort (Supplementary Table 4, both *P* < 0.05). Taken collectively, genomic alterations of potentially targetable genes (top 5) occurred in ~74.1% of the entire AFPGC cohort (Fig. 4b).

Noting that *ERBB2* and *CCNE1* were two well-known oncogenes in 17q12 and 19q12, respectively, the status of *ERBB2* and *CCNE1* was examined by FISH or IHC in a larger clinical cohort (*N* = 105, Fig. 4c). The *ERBB2*-positive rate in AFPGC was significantly higher than that in TCGA-GC and subtypes (Fig. 4d and Supplementary Table 5). A similar trend was also observed with regard to *CCNE1*. Furthermore, the prognostic value of *ERBB2* and *CCNE1* in 105 AFPGC patients was explored. The results revealed that the patients with a positive status of *ERBB2* or *CCNE1* had worse survival outcomes when compared to those with a negative status in AFPGC (HR = 2.07, *P* = 0.009, Fig. 4e; HR = 2.64, *P* < 0.001, Fig. 4g). However, such phenomenon was not found in the TCGA-CIN cohort, which harbors the most frequent ERBB2 and CCNE1 amplification in TCGA subtypes (Fig. 4f, h). Moreover, the associations of *ERBB2* and *CCNE1* with clinicopathological features in AFPGC were evaluated (Supplementary Table 6). The positive status of *ERBB2* showed an association with advanced tumor-node-metastasis (TNM) stage

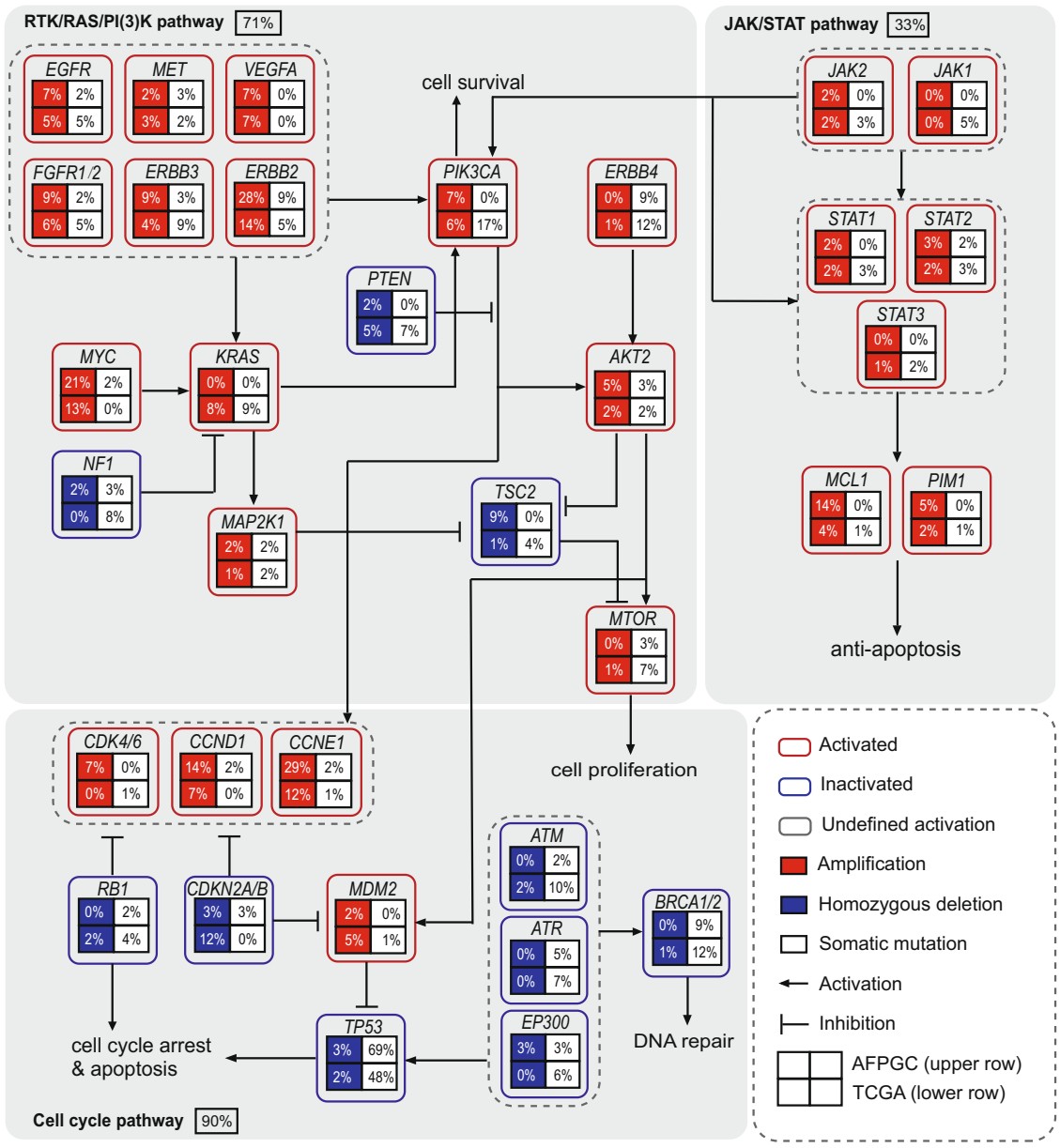

**Fig. 3 Comparison of genetic alterations of signaling pathways in AFPGC and TCGA-GC.** Somatic mutations and SCNAs in key signaling pathways are shown for AFPGC ($n = 58$) and TCGA-GC ($n = 393$) cohort. Three major signaling pathways, namely RTK/RAS/PI(3)K pathway, cell cycle pathway, and JAK/STAT pathway, were shown. In a square grid, the upper row represents AFPGC and the lower row represents TCGA-GC. Percentages represent the alteration frequencies. SCNAs, somatic copy number alterations. Source data are provided as a Source Data file.

($P = 0.035$), positive lymphovascular invasion ($P = 0.033$), and high liver metastasis ratio ($P = 0.025$). Similarly, patients with a positive status of *CCNE1* demonstrated higher ratio of lymphovascular invasion ($P = 0.012$) and liver metastasis ($P = 0.003$). These findings indicated that amplification in *ERBB2* or *CCNE1* might contribute to tumor progression in AFPGC. To examine the combined effects of *ERBB2* and *CCNE1* on prognosis, the patients ($n = 105$) were divided into three groups according to the status of *ERBB2* and *CCNE1* (Co-positive group: *ERBB2* positive and *CCNE1* positive; Single positive group: *ERBB2* positive or *CCNE1* positive; and Co-negative group: *ERBB2* negative and *CCNE1* negative). Interestingly, the co-positive group demonstrated the worst OS rate and aggressive biological characteristics (Fig. 4i and Supplementary Table 7). Although *ERBB2* and *CCNE1* amplification were also enriched in TCGA-CIN, the combined effects of *ERBB2* and *CCNE1* on prognosis were not observed (Fig. 4j).

Furthermore, multivariate Cox regression analysis demonstrated a positive status of *ERBB2* or *CCNE1* was an independent prognostic factor in AFPGC (after adjusting by *TP53* mutation status) but not in TCGA-CIN (Fig. 4k, l).

**Validation of potential therapeutic targets in PDX models.** A series of passable PDX models of AFPGC were successfully established. The status of AFP, *ERBB2*, and *CCNE1* were confirmed by FISH and/or IHC (Fig. 5a and Supplementary Fig. 9) and Fig. 5b–d showed the representatives of AFPGC PDX models with different status of *ERBB2* and *CCNE1*. The clinicopathological features and detailed alterations for each PDX model are summarized in Supplementary Table 8 and Supplementary Fig. 10. *CCNE1*, along with its catalytic subunit Cyclin-dependent kinase 2 (CDK2), plays a critical role in the

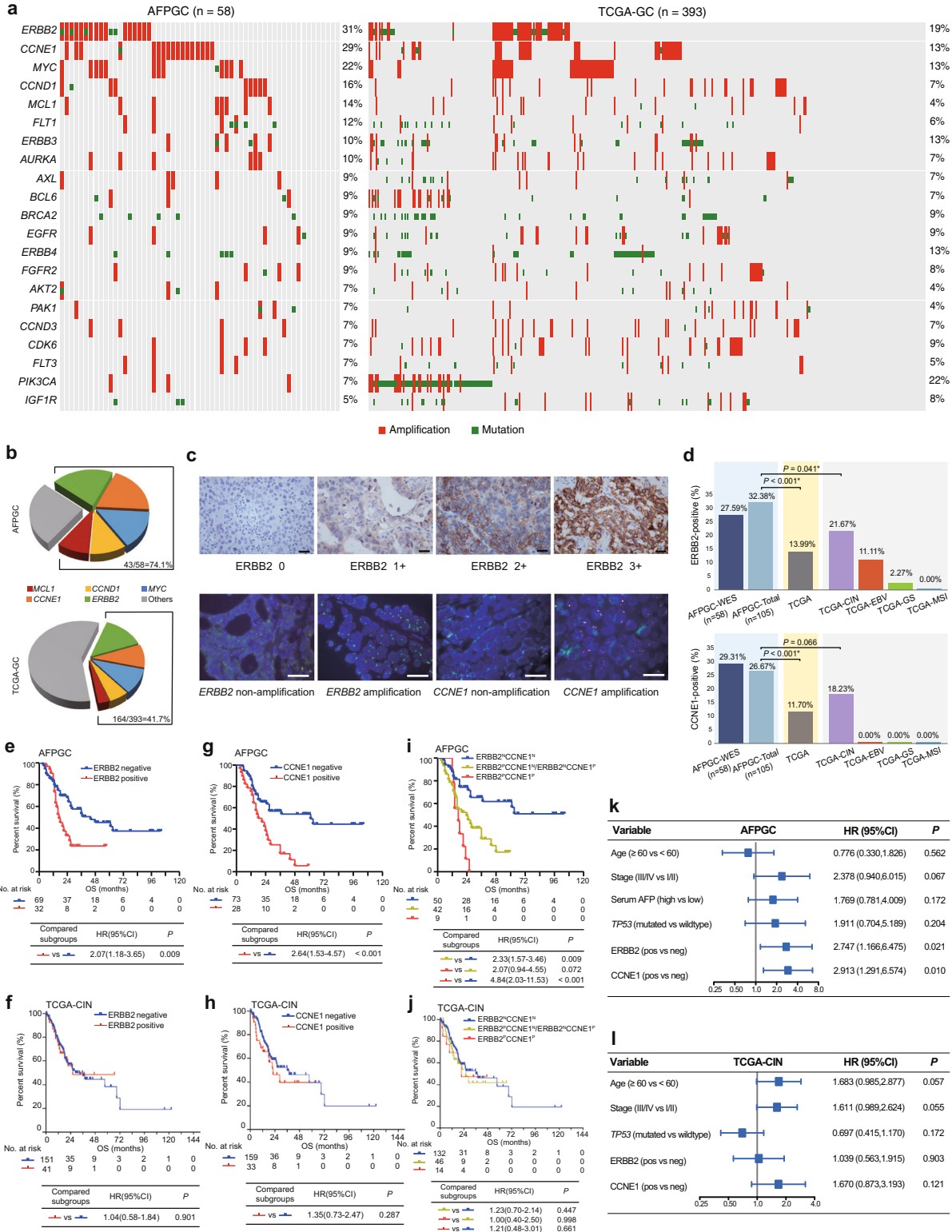

regulation of G1–S phase transition. Targeting CDK2 has been validated to be an effective anticancer strategy for *CCNE1*-amplified cancer[16,17]. The efficacy of AZD5438 (inhibitor of CDKs, including CDK2) in five AFPGC PDX models with or without *CCNE1* amplification was evaluated. The results showed statistically significant differences in tumor volume between the control group and the AZD5438-treated group in *CCNE1*-amplified AFPGC PDX models, but not in non-*CCNE1*-amplified models (Fig. 5e–h). Next, the antitumor activity of trastuzumab

was evaluated. Consistent with the previously published report, trastuzumab exerted antitumor effects in ERBB2-positive AFPGC PDX models (Fig. 5i, j). Furthermore, the antitumor activity of AZD5438 and trastuzumab was evaluated on ERBB2-positive PDX AFPGC models. According to the previous study, *CCNE1* amplification or overexpression participated in the resistance to trastuzumab treatment in breast cancer[18]. Interestingly, statistically significant differences were also present in the tumor volume between the group treated with AZD5438 combined with

**Fig. 4 Potentially targetable alterations in AFPGC and TCGA-GC, and survival analysis. a** Genomic alterations of potentially targetable genes in AFPGC and TCGA-GC. **b** Overall frequency of selected targetable genes in AFPGC and TCGA-GC. **c** Representative images of *ERBB2* and *CCNE1* status. Presented data are a representative image of three independent experiments. Upper panel shows IHC score of *ERBB2* with 0, 1+, 2+, and 3+. Lower panel shows representative FISH image of *ERBB2* and *CCNE1*. Scale bar represents 50 μm. **d** The positive rate of *ERBB2* and *CCNE1* in AFPGC and TCGA-GC. **e, f** Associations between *ERBB2* status and OS in AFPGC (**e**) or in TCGA-CIN (**f**). **g, h** Associations between *CCNE1* status and OS in AFPGC (**g**) or in TCGA-CIN (**h**). **i, j** Associations between combined status of *ERBB2* and *CCNE1* and OS in AFPGC (**i**) or in TCGA-CIN (**j**). **k, l** Forest plot of multivariable Cox proportional hazard regression in AFPGC (**k**) or in TCGA-CIN (**l**). The hazard ratios are presented and the horizontal lines indicate the 95% confidence intervals. In AFPGC-WES and TCGA-GC, the positive status of ERBB2/CCNE1 was defined as amplification by GISTIC 2.0. In AFPGC-total, the definition of the positive status of ERBB2 and CCNE1 was shown in Supplementary Methods. Statistical significance was determined using two-sided $\chi^2$-test (**d**), log-rank (Mantel–Cox) test (**e–j**), and multivariate Cox regression (**k, l**). AFPGC-WES, the AFPGC cohort with WES data; AFPGC-total, the whole AFPGC cohort in this study; $CCNE1^N$, *CCNE1* negative; $CCNE1^P$, *CCNE1* positive; CI, confidence interval; $ERBB2^N$, *ERBB2* negative; $ERBB2^P$, *ERBB2* positive; FISH, fluorescence in situ hybridization; HR, hazard ratio; IHC, immunohistochemistry; neg, negative; OS, overall survival; pos, positive. Source data are provided as a Source Data file.

trastuzumab and those treated with AZD5438 or trastuzumab alone in *ERBB2* and *CCNE1* co-amplified PDX models, but the results were not observed in *CCNE1* non-amplified PDX models (Fig. 5i–k).

The treatment efficiency was further evaluated by western blotting analysis in ERBB2 and CCNE1 co-positive PDX model after drug administration (Fig. 5l). The results showed that AZD5438 exerted an antitumor effect by inhibiting the expression of CDK2 and phosphorylated retinoblastoma (p-Rb). Moreover, the expressions of p-ERK, p-AKT, CDK2, and p-Rb were greatly decreased in the combined trastuzumab and AZD5438 treatment group as compared to other groups.

## Discussion

In this study, we provided an understanding of the clinicopathological and molecular features of AFPGC and focused on targetable genomic alterations. We validated that AFPGC had a higher liver metastasis rate and poorer prognosis. Frequent genetic alterations, as well as key signaling pathways that might contribute to the tumorigenesis and development of AFPGC, were revealed. Also, the potentially targetable genes in AFPGC were analyzed and translational research was performed to explore the precise target therapy on PDX models for this disease.

AFPGC was first described by Bourrille et al.[19] in 1971 and was characterized as a special subtype of GC according to the WHO classification of digestive system tumors. Histologically, AFPGC can be identified as hepatoid adenocarcinoma[20] and other differentiated adenocarcinomas, such as tubular/papillary adenocarcinoma, enteroblastic adenocarcinoma, and yolk-sac tumor-like carcinoma[21–25]. More than one of these histological types often coexists in AFPGC.

A study conducted by Hirajima et al.[4] has indicated liver metastasis as a dominant character (which occurred in nearly 40% of AFPGC), as well as an independent prognostic factor in AFPGC[4]. Liu et al.[5,26] have conducted a study on a clinical cohort of 104 AFPGC patients, which showed a high incidence of liver metastasis rate (60%) and dismal prognosis. Similarly, a liver metastasis rate of 40.1% was found in our cohort of 105 AFPGC patients. In addition, the prognosis of AFPGC patients was poorer than those with stage-matched non-AFPGC. However, so far there is no specialized treatment for AFPGC either in the current Japanese Gastric Cancer Association guidelines or National Comprehensive Cancer Network guidelines[27,28]. Till date, very little is known regarding the molecular characteristics of AFPGC that seriously restricted the understanding and management of this disease.

Based on WES results in our study, the significantly mutated genes of AFPGC, such as *TP53*, *CSMD3*, *FPR1*, *PCLO*, *KMT2C*, *SYNE1*, and *LRP1B* were shown. *TP53* acts as a key tumor suppressor that plays a major role in preserving genomic stability[29].

*TP53* mutations are associated with increased amplification of oncogenes, deep deletions of tumor suppressor genes, and nucleotide-level mutation rates. In our analysis, *TP53* was more frequently mutated in AFPGC when compared to TCGA-GC (most of the cases were negative for AFP[30]). Despite the similarity of the mutation rate, the distribution of *TP53* mutations in AFPGC was different from TCGA-CIN. For instance, three cases in AFPGC harbored X331_splice mutation, which was absent in TCGA-GC but present in other cancer types including lung, colon, and liver cancers[31]. This mutation was predicted as "disease causing" by MutationTaster[32] and may induce loss of capacity for oligomerization, leading to partial or complete loss of transactivation potential of p53[33]. Notably, as the most frequent hotspot in TCGA-GC (or TCGA-CIN), TP53 R175 mutation is absent in AFPGC. This mutation was reported to reduce the DNA-binding capacity of p53 (loss-of-function) and was oncogenic in multiple cancers including GC[34]. Previous analysis on TCGA pan-cancer database involving 10,225 samples from 32 cancers also revealed that *TP53* mutations were associated with poorer prognosis in human cancer[35]. In this study, *TP53* mutations were associated with poorer prognosis of AFPGC patients using Kaplan–Meier analysis (Supplementary Fig. 11), but not an independent prognostic factor. CUB and sushi multiple domains (*CSMD3*) was shown to participate in dendrite development[36]. Previous studies have reported that *CSMD3* is involved in tumorigenesis and tumor proliferation in lung cancer[37] and liver cancer[38]. Moreover, *CSMD3* acts as an immune regulator and might be associated with immune response[39,40]. The formyl peptide receptor 1 (*FPR1*) is one of the members of the formyl peptide receptor family, which is involved in cell motility, angiogenesis, inflammation, and immune response[41–43]. *FPR1* showed association with epithelial mesenchymal transition, proliferation, and migration in different cancer types[44,45]. In this study, *FPR1* tended to show mutations more frequently in AFPGC than TCGA-GC or TCGA-CIN. Moreover, *FPR1*-mutated-AFPGC patients tended to have a higher distant metastatic rate when compared to *FPR1*-wild-type patients (80% vs. 38%, $P = 0.149$). A recent study demonstrated that *FPR1* played a key role in chemotherapy-induced anticancer immune response[40]. In our analysis, we found that the frequency of *KMT2C* mutations in AFPGC was significantly higher than that in TCGA-CIN. Previous studies reported that Lysine (K)-specific methyltransferase 2C (*KMT2C*), encoding an H3K4 histone methyltransferase, was crucial for tumorigenesis and progression in various cancer[46,47]. Notably, a recent study demonstrated that *KMT2C* mutations promoted epithelial-to-mesenchymal transition in gastric adenocarcinoma[48]. Apart from significantly mutated genes, we also compared the frequently mutated genes (>10%) between AFPGC and TCGA-GC/TCGA-CIN (Supplementary Data 9 and 10). Understanding the role of key mutations in tumorigenesis,

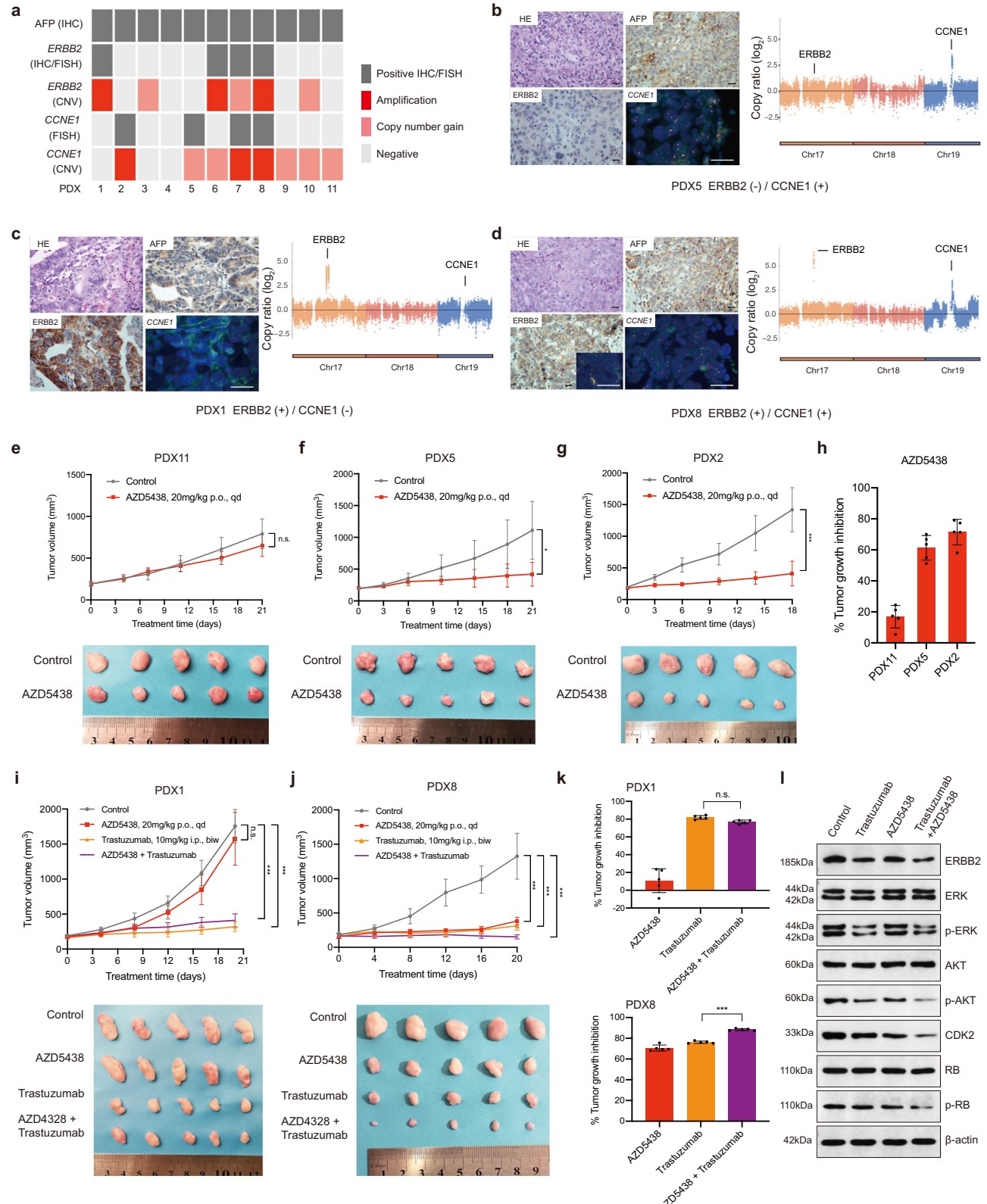

tumor progression, and metastasis are important areas of AFPGC research in the future.

SCNA is defined as DNA segments (~1 kb or larger) that showed copy number differences among haplotypes[49]. SCNAs were responsible for evolution, genetic diversity, and genomic disorders including cancers[50,51]. So far, previous reports on SCNA analysis of AFPGC were limited. Wang et al.[52] have reported that the frequent copy number gains at 20q11.21-13.12 in hepatoid adenocarcinoma of the stomach (a differentiated type of AFPGC) tended to be related to more adverse biobehaviour, including poorer differentiation, greater vascular and nerve invasion, and greater liver metastasis, but showed no statistically significant differences. In this study, we detected frequent copy number amplification alterations containing, e.g., *CCNE1*

**Fig. 5 Validation of selected genes as potential therapeutic targets in PDX models of AFPGC. a** The status of AFP, ERBB2, and CCNE1 in AFPGC PDX models. The CNVs status of ERBB2 and CCNE1 are defined by the WES results of corresponding patients. **b–d** The representatives of AFPGC PDX models with different status of ERBB2 and CCNE1. Scale bar represents 50 μm. Data were obtained from three independent experiments. The right panel represents the CNVs status of ERBB2 and CCNE1 in the corresponding patients. **e–h** In vivo sensitivity of AFPGC with or without CCNE1 amplification to CCNE1 inhibition (AZD5438) (n = 5 biologically independent samples). Tumor volumes and proportion of tumor growth inhibition were shown as means ± SD. **i–k** In vivo sensitivity of ERBB2-positive AFPGC with or without CCNE1 amplification to ERBB2 (Trastuzumab) and CCNE1 (AZD5438) dual inhibition (n = 5 biologically independent samples). Tumor volumes and proportion of tumor growth inhibition were shown as means ± SD. **l** The western blot analysis of critical molecules (ERBB2, ERK, p-ERK, AKT, p-AKT, CDK2, RB, and p-RB) for ERBB2 and CCNE1 co-positive PDX models (PDX8) were performed. The western blotting images are representative of three different independent experiments. Statistical significance was determined using two-sided Student's t-test (**e–g**) and one-way ANOVA (**i–k**). biw, twice a week; Chr, chromosome; CNVs, copy number variations; FISH, fluorescence in situ hybridization; HE, hematoxylin–eosin staining; IHC, immunohistochemistry; i.p, intraperitoneal injection; kD, kilodaltion; p.o, oral gavage; qd, per day. *P < 0.05, **P < 0.01, ***P < 0.001; n.s., not significant. Source data are provided as a Source Data file.

(19q12), ERBB2 (17q12), CCND1 (11q13.3), MUC4 (3q29), and MCL1 (1q21.3).

In this study, we found that AFPGC presented with recurrent CNAs. Considering chromosomal instability (CIN) is a hallmark for TCGA-CIN[8,53], we, therefore, compared the significant SCNA profiles between AFPGC and TCGA-CIN. Nearly 57% statistically significant amplification regions and 73% deletion regions were found in AFPGC specifically. These SCNA regions in AFPGC contain important cancer-related genes, such as TRIM28 (19q13.43), which has been proved to accelerate cell proliferation and metastasis in a variety of human cancer[54], and FAT1 (1p36.32), which was observed in multiple human cancers to promote Wnt/β-catenin signaling and tumorigenesis[55]. CIN is a common feature of gastrointestinal adenocarcinomas (GIACs)[8,53,56] and nearly 70% of GIACs can be classified into CIN subtype[57]. Recent studies further classified CIN subtype of GC as well as other GIAC into subclasses with different genomic alterations and clinical significance[57,58], suggesting that CIN subtype of GC may still be a heterogeneous group. A recent study also demonstrated that CIN is a driver of metastatic progression[59]. This might partially contribute to the aggressive phenotype of AFPGC. In addition, Bakhoum et al. observed that metastasis from CIN-high tumor cells tended to involved multiple organs[59]. Similarly, we also found that CIN subtype of GC displayed a multi-organ metastatic pattern by analyzing an MSK cohort reported by Janjigian et al.[60]. However, AFPGC showed a dominant tendency of liver metastasis (Supplementary Fig. 12), suggesting the different metastatic patterns between AFPGC and CIN subtype of GC. In all, the role of CIN in AFPGC is interesting and worth further exploration in larger cohorts by multi-omics in the future.

Combined with WES and clinical information, AFPGC patients with amplifications at 17q12 or 19q12 showed significantly worse survival than those without amplification. It is noteworthy that ERBB2 and CCNE1 have been proposed as two well-known oncogenes at 17q12 and 19q12, respectively[61–63]. ERBB2 acts as a critical therapeutic target and targeting ERBB2 in the treatment of ERBB2-positive metastatic GC has proven to be an effective therapeutic strategy[64]. CCNE1 amplification is associated with poorer survival rate in patients with different types of cancers[65,66] and has been reported to associate with liver metastasis in TP53-mutated GCs[67]. The frequency of ERBB2 and CCNE1 amplification were 32.4% and 26.7% in our large cohort of AFPGC. Also, we demonstrated that both ERBB2 and CCNE1 amplification showed significant correlation with poor OS and several aggressive clinicopathological characteristics of AFPGC. This study reported that patients who harbored both ERBB2 and CCNE1 amplification demonstrated the worst prognosis in AFPGC. Furthermore, multivariate Cox regression analysis demonstrated that a positive status of ERBB2 or CCNE1 was an independent prognostic factor in AFPGC; however, this prognostic value was not observed in TCGA-CIN. ACRG-MSS/TP53− is another subtype of GC classification reported by Cristescu

et al.[9] and enriches ERBB2 and CCNE1 amplification. The prognostic value of ERBB2 and CCNE1 amplification in ACRG-MSS/TP53− was distinct from that in AFPGC (Supplementary Fig. 13). These findings implied the important role of ERBB2 and CCNE1 amplification in the development and progression of AFPGC.

To further explore the translational significance of the above findings, a series of AFPGC PDX models were established in our laboratory. Previous studies have shown that PDXs could well maintain the principal histological and genetical characteristics of their donor tumors and remain stable across the passages[68,69]. Our research group has previously applied GC PDX as a platform for evaluating drug sensitivity[70,71]. In this study, a large collection of AFPGC PDX models was reported. ERBB2 and CCNE1 as two frequently altered genes were taken as a translational example to explore the precise therapeutic target for this disease. Cyclin E1 (CCNE1), along with its catalytic subunit CDK2, plays a critical role in regulating the G1–S phase transition[72]. It has been reported that CCNE1 amplification acts as a positive marker of CDK2 inhibitor sensitivity in breast as well as ovarian cancers[18,73]. AZD5438, a CDK1/2/9 inhibitor, exerts a significant antitumor effect in previous studies[17,74]. In our cohort, nearly 30% of AFPGCs harbored CCNE1 amplification. Our vivo study revealed that AZD5438 showed significant antitumor activity in CCNE1-amplified AFPGC PDX models. In our study, AFPGC patients involved a higher proportion of ERBB2-positive cases (32.4%) when compared with common GC (12–20%)[75]. Also, the ERBB2-positive GC PDX showed a good response to trastuzumab, which was consistent with the previous studies[64,76]. In our study, nearly 20% of ERBB2-positive AFPGC was accompanied by CCNE1 amplification. Interestingly, our data showed that the co-positive of ERBB2 and CCNE1 were correlated with more aggressive behavior and poorer prognosis than single positive or dual negative. Previous studies have shown that the co-amplification of CCNE1 and ERBB2 is associated with trastuzumab resistance in breast cancer and GC[64,76]. Moreover, it has been reported that CDK2 inhibitor could augment the anti-proliferative impact of lapatinib (ERBB2 inhibitor) in CCNE1/ERBB2 co-amplified GC cells[77]. Therefore, we hypothesized that the dual-targeted therapy combination of ERBB2 and CCNE1 for AFPGC could be an effective therapeutic strategy to overcome or reverse the potential treatment resistance and have a synergistic antitumor effect. Encouragingly, AZD5438 was able to augment the antitumor effects of trastuzumab in AFPGC PDX model with co-amplification of ERBB2/CCNE1. Our results initially suggested that dual inhibition of ERBB2 and CCNE1 acts as a potential treatment strategy for AFPGCs with ERBB2/CCNE1 co-amplification. Besides CCNE1 and ERBB2, our study also provided a list of other potential therapeutic targets, deserving further exploration and verification. However, clinical trials investigating the above potential targets are unlikely to occur.

PDX model in our study plays an important platform for translational research and animal clinical trials for AFPGC.

There are some limitations that require acknowledgement in our study. First, in order to efficiently select AFPGCs from more than 5200 GC patients, we set relatively strict inclusion criteria, which may rise to selection bias to some extent. Although this study represented large genomic analyses of AFPGC samples, our analyses were still limited by sample size. That is, the results, such as significantly mutated genes, chromosomal alterations, and the correlation between genetic alterations and clinicopathological variables (e.g., histopathology), should still be fully validated in a larger multi-centered cohort. Because of the retrospective study design, the sample quality could not meet the criteria of multiple omics sequencing. Moreover, limited by the delayed approval of trastuzumab by the National Medical Products Administration of China and relatively low affordability of Chinese patients, the proportion of patients who received trastuzumab treatment in this study was relatively low, leading to insufficient cases to evaluate the efficacy of ERBB2-targeted therapy. PDX was an effective technology to explore therapeutic strategies. The effect of targeting therapeutic genomic alterations was evaluated in limited AFPGC PDXs due to scarcity of the disease and low success rate to generate and recover PDXs. More PDX samples can enhance the clinical implication of this study. Thus, a clinically reliable platform with more AFPGC PDX models should be developed and more clinical trials should be conducted in the future.

In summary, our study presented a large genomic landscape of AFPGC so far, which is a critical step in our efforts for understanding this disease. Besides, a series of PDX models were generated, which could be a platform to identify potential molecular targets and optimize therapeutic approaches for AFPGC.

## Methods

**Patients and samples.** From January 2011 to December 2018, there were 121 cases of primary gastric carcinoma with elevated serum AFP levels (≥20 ng/ml) at diagnosis among 5261 GC cases at the First Affiliated Hospital, School of Medicine, Zhejiang University. One hundred and five patients with AFP-positive IHC staining were finally enrolled. For comparison, we randomly selected 1 : 3 cases (311 patients) with stage-matched primary GCs with normal serum AFP levels in our institution. We retrospectively collected fresh-frozen or FFPE tumor tissues and matched tumor adjacent normal tissues from 58 AFPGC patients for genomic characterization, after excluding 47 patients with prior chemotherapy (23 patients), insufficient tumor volume (12 patients), and low quality of DNA (12 patients). Staging was performed according to the American Joint Committee on Cancer TNM Staging Classification for Carcinoma of the Stomach (8th edition, 2018). Follow-up data were obtained by phone, letter, and the out-patient clinical database (last follow-up was September 2019) and follow-up information were available in 101 (101/105, 96.2%) patients. The OS time was calculated from the date of diagnosis to the last day of follow-up or the date of death. Patient-derived paraffin-embedded tissue samples were used in accordance with ethical guidelines in the First Affiliated Hospital, School of Medicine, Zhejiang University (No. 2018-309). Patient-written consents were obtained from study participants, informing the use for genomic sequencing, drug test, and publication.

**DNA extraction and DNA quantification and qualification.** For fresh-frozen samples, genomic DNA from tumor tissues and matched normal tissues were isolated using the QIAamp DNA Mini Kit (Qiagen) according to the manufacturer's instructions. Further, for FFPE tumor tissues, DNA was extracted using QIAamp DNA FFPE Tissue Kit (Qiagen). Then we combined the following two methods to verify the quality of isolated genomic DNA. First, DNA degradation and contamination were monitored on 1% agarose gels. Second, Qubit® DNA Assay Kit in Qubit® 2.0 Flurometer (Invitrogen, USA) was used to quantify DNA concentration.

**WES and data analysis.** Whole-exome library construction was generated using the Agilent SureSelect Human All Exon V6 Kit (Agilent Technologies, Santa Clara, CA, USA). According to the manufacturer's instructions, the index-coded samples were clustered on a cBot Cluster Generation System using Hiseq PE Cluster Kit (Illumina). After cluster generation, the DNA libraries were sequenced on Illumina Hiseq platform (Illumina, San Diego, California, USA) and 150 bp paired-end reads were generated.

Data quality control was first performed and all downstream bioinformatics analyses were based on high-quality clean data, in which reads containing an adapter, reads containing poly-N, and low-quality reads were removed. The paired-end clean reads were aligned to the Human Genome Reference Consortium build 37 (GRCh37) using BWA v.0.7.8[78].

**Somatic mutation detection and significantly mutated genes identification.** Identification of somatic SNVs was conducted by muTect[79] and the somatic InDels were detected by Strelka[80]. Paired adjacent normal tissue DNA was obtained as a control for all tumor samples. To reduce false-positive calls in FFPE specimens, only mutation site with a minimum of three variant reads and a variant allele frequency > 0.08 was used for further analysis. ANNOVAR (ANNOVAR_2015Mar22)[81] was used to annotate variant call format files. The MuSiC algorithm[13] was used to identify significantly mutated genes from the profiles of somatic SNVs and InDels in AFPGC (false discovery rate (FDR) < 0.25).

**Mutational signatures analysis.** There are six variant types of single base substitution as follows: C>A/G>T, C>G/G>C, C>T/G>A, T>A/A>T, T>C/A>G, and T>G/A>C. According to the number of point mutations of different types, we conducted a cluster analysis to observe the similarity and difference within tumor samples. Considering the type of base at 1 bp in the upstream and downstream of the point mutations, the point mutation was divided into 96 types. According to the frequency of 96 mutation types, the point mutation types were decomposed into several different mutation characteristics by non-negative matrix factorization method[82]. Then, extracted mutational signatures were compared to the pan-cancer catalog of 30 known signatures referenced in the Catalogue of Somatic Mutations in Cancer (COSMIC) database using SomaticSignatures packages[83]. The similarity of mutation signatures was evaluated with cosnine similarity > 0.85, which suggests common signatures. Signatures 1, 17, and 29 were identified in our samples. There were two signatures common to signature 1 and we kept the higher value of cosnine similarity signature as signature 1.

**Copy number analysis.** We identified SCNAs using CNVkit[84] to analyze the copy number state of each tumor. Then we used GISTIC 2.0[85] to evaluate the genome regions with significant amplification or deletion in the samples and screen out the regions with high frequencies, namely recurrent CNA regions. The GISTIC score reflects the frequency of CNA in this segment. Red and blue indicate an increase and a decrease in the number of copies, respectively. Then we downloaded TCGA-GC copy number data from TCGA and identified significant SCNAs using GISTIC, to compare AFPGC SCNA profiles with TCGA-GC and subtypes (TCGA-CIN, TCGA-EBV, TCGA-GS, and TCGA-MSI).

**Integrated pathway analysis.** We downloaded the geneset of cancer-driving genes from Cancer Gene Census in the COSMIC, then investigated the genetic alteration rate of the cancer-driving genes in AFPGC. Subsequently, we screened out the frequently altered genes and performed KEGG pathway enrichment using DAVID bioinformatics Resource 6.8. We found several key pathways significantly enriched in AFPGC (FDR < 0.05). Besides, we downloaded TCGA-GC data from TCGA and compared genetic alterations (somatic mutations and SCNAs) of the key pathways in AFPGC and TCGA-GC.

**Analysis of potential targetable genes.** We screened out a list of genes altered at least in four cases in our AFPGC cohort that are potentially targetable (based on COSMIC, OncoKB databases[86], MSK-IMPACT[87], and comprehensive literature review). These genetic alterations of targetable genes were compared between AFPGC and TCGA-GC, then visualized by ComplexHeatmap package[88].

**H&E and IHC staining.** Tumor tissues were fixed in 10% neutral buffered formalin followed by gradient dehydration, wax immersion, embedding, and sectioning, then evaluated by hematoxylin and eosin (H&E) and IHC staining. For H&E staining, sections were processed by dewaxing, hydration, dyeing, dehydration, and sealing. For IHC staining, sections were dewaxed, rehydrated, and subjected to antigen retrieval. Then we incubated the sections with primary antibodies against ERBB2 (1 : 200, Cell Signaling Technology, #2165) and AFP (1 : 100, ProteinTech, #14550-1-AP) at 4 °C for 12 h after quenching endogenous peroxidase activity and blocking nonspecific binding sites. This was followed by a 30 min incubation with secondary antibody [SP Rabbit & Mouse HRP Kit (DAB) (1 : 400, CoWin Biosciences, #CW2069)]. We performed IHC using the streptavidin–biotin peroxidase complex method (Lab Vision, Fremont, CA, USA), then observed and photographed the sections using an optical microscope (Nikon, Tokyo Japan). IHC results were evaluated according to previously published methods[64,89] and details were shown in Supplementary Methods. ERBB2-positive status included ERBB2 overexpression (score 3+) by IHC or ERBB2 amplification by FISH as described in previous studies[64,90].

**Fluorescence in situ hybridization.** FISH analysis was performed for ERBB2 and CCNE1 gene assessment on FFPE tumor tissues using ERBB2/CEN17 dual color Probe (ZytoVision, #Z-2020-20) and CCNE1/Con19 FISH Probe (Empire

Genomics, #*CCNE1*-CHR19-20-ORGR) following the manufacturers' recommendations. Probes were co-denatured for 10 min at 75 °C (ZytoVision)/3 min at 83 °C (Empire Genomics) on the slide and incubated overnight at 37 °C. Three post-hybridization washes were performed in 1× Wash Buffer A at 37 °C for 5 min each. Finally, the slides were air dried and counterstained with 4′,6-diamidino-2-phenylindole/antifade solution. Signals for each locus-specific FISH probe were evaluated under an Olympus BX51 microscope (Olympus Corporation, Tokyo, Japan). *CCNE1* gene amplification results were based on at least 50 evaluable tumor nuclei. *CCNE1* amplification was defined as the presence of either loose or tight *CCNE1* cluster, or the ratio of *CCNE1/Con19* ≥ 2.0[91]. *ERBB2* gene status was classified as non-amplified (*ERBB2/CEN17* ratio < 2.0) or amplified (*ERBB2/CEN17* ratio ≥ 2.0)[64,92]. Once an adequate target area had been identified, the scores for *ERBB2* and *CEN17* copy numbers present in 20 representative nuclei were scored. If the resulting ratio of *ERBB2/CEN17* fell within 1.8–2.2, an additional 20 nuclei were scored and the resulting ratio was calculated from a total of 40 nuclei.

**Western blotting**. Lysates of frozen tumor tissue were extracted by lysis buffer containing protease and phosphatase inhibitors (Sigma-Aldrich, St. Louis, MO, USA), then fractionated by SDS-polyacrylamide gel electrophoresis and electro-transferred to polyvinylidene difluoride membranes (Millipore, Billerica, MA, USA). After blocking in 5% skim milk in triethanolamine buffered saline- tween for 1 h at room temperature, the membranes were incubated with primary antibodies overnight at 4 °C. The membranes were washed and then incubated with peroxidase-conjugated secondary antibodies [HRP Conjugated AffiniPure Goat Anti-rabbit IgG (H + L) (1 : 5000, BOSTER, #BA1054) and HRP Conjugated AffiniPure Goat Anti-mouse IgG (H + L) (1 : 5000, BOSTER, #BA1051)] for 1 h at room temperature. After washing three times with triethanolamine buffered saline, the blot was soaked in ECL™ chemiluminescent detection reagents (Millipore, Billerica, MA, USA) for 1 min. The membranes were then exposed to film (Kodak, Rochester, NY, USA) for 30 s in a darkroom. The antibodies against the following proteins were used: ERBB2 (1 : 1000, Cell Signal, #2165), AKT (1 : 1000, Cell Signal, #4691), p-AKT (1 : 1000, Cell Signal, #13038), ERK (1 : 1000, Cell Signal, #4695), p-ERK (1 : 2000, Cell Signal, #4370), CDK2 (1 : 3000, ProteinTech, 10122-1-AP), Rb (1 : 1000, ProteinTech, 17218-1-AP), p-Rb (1 : 1000, Cell Signal, #8516), and β-Actin (1 : 500, BOSTER, #BM0627).

**Establishing xenografts and treatment protocol**. Four- to six-week-old female BALB/c nude mice, purchased from Shanghai Slac Laboratory Animal Corporation (Shanghai, China), were housed with regular 12 h light/12 h dark cycles for at least 3 days before use. Ambient temperature was 20 ~ 22 °C, kept at constant humidity of 40 ~ 60%. PDX models were established as we previously reported[71,93]. We monitored xenograft growth at least twice weekly by a vernier caliper measuring the length (*L*) and width (*W*) of the tumor, and then removed them for serial transplantation after the volume reached about 1500 mm³. The tumor volume (V) was calculated according to the following formula: $V = L \times W^2/2$.

We used xenografts from the third generation (the second mouse-to-mouse passage) for the experiments once the tumor volume reached about 150–200 mm³. Mice were randomly assigned to different groups (five to seven mice/group)as follows: (i) vehicle; (ii) Trastuzumab (SelleckChem) 10 mg/kg twice weekly of intraperitoneal injection; (iii) AZD5438 (SelleckChem) 20 mg/kg daily by oral gavage; (iv) AZD5438 + Trastuzumab, for 3 weeks. Experiments were ended once the tumor volume surpassed 1500 mm³ or mouse weight loss reached 20%. The percentage of tumor growth inhibition (TGI) was calculated according to the following formula: $TGI = (1 - T/C) \times 100\%$, where $T/C$ represents the relative tumor volume of treatment group and control group. After the mice had been killed, we conducted immunoblot to assess the expression of various markers. Animal care and experiments were performed under the approval and supervision of the Animal Experimental Ethical Inspection of the First Affiliated Hospital, College of Medicine, Zhejiang University (number 2018-378).

**Statistical analysis**. Two or multiple continuous variables were compared using Student's *t*-test or one-way analysis of variance, respectively. The $\chi^2$ and Fisher's exact tests were used to analyze the significance of categorical data. Survival rates were calculated by the Kaplan–Meier method and compared using the log-rank test. Multivariate Cox regression analysis was performed to identify the independent prognostic factors. Non-parametric variables were compared using the Kruskal–Wallis test and the Mann–Whitney test. Statistical analysis was performed using SPSS 21.0 software. $P < 0.05$ was considered statistically significant.

**Reporting summary**. Further information on research design is available in the Nature Research Reporting Summary linked to this article.

## Data availability

The raw sequence data reported in this study have been deposited in the Genome Sequence Archive[94] in National Genomics Data Center[95], Beijing Institute of Genomics (China National Center for Bioinformation), Chinese Academy of Sciences [http://bigd.big.ac.cn/gsa-human/] (accession number: HRA000429). The whole-exome somatic variants are also publicly available from the European Nucleotide Archive [https://www.ebi.ac.uk/ena/browser/home] (study ID: PRJEB44858 (ERP128950)). The clinical data are provided in Supplementary Data 1. A complete list of TCGA cohort can be found in Supplementary Data 11. The MSKCC data are available in the cBioPortal for Cancer Genomics database [https://www.cbioportal.org/study/summary?id=egc_msk_2017]. The TCGA data are available in the cBioPortal for Cancer Genomics database [https://www.cbioportal.org/study/summary?id=stad_tcga]. TGCA data analyzed for this manuscript were released on 28 January 2016. All other data are available within the Article and Supplementary Information. Source data are provided with this paper.

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

## Acknowledgements

We thank Zheng Ju and Hong Su (from Novogene Bioinformatics Technology Co. Ltd, Beijing, China) for assistance with bioinformatics analyses. We greatly thank all the staff and assistants in the Department of Surgical Oncology, the First Affiliated Hospital, Zhejiang University School of Medicine, for their support in collecting samples. This project was supported by National Natural Science Foundation of China (Grant number 81803107), Zhejiang Provincial Natural Science Foundation of China (LQ18H160002 and LQ20H160043), the Key Research and Development Program of Science and Technology Department of Zhejiang Province (2018C03022), Zhejiang Provincial Department of Science and Technology (number 2020C03112), and Project of the Regional Diagnosis and Treatment Centre of the Health Planning Committee (number JBZX-201903).

## Author contributions

L.T., Haiyong Wang, Z.Z., J.L, Y.D., and Yanyan Chen are responsible for the study concept and design. J.L., Y.D., J.J., and Yiran Chen are responsible for the development of methodology. Yanyan Chen, J.J., Yiran Chen, Y.H., M.W., C.L., and M.K. are responsible for the acquisition of data. J.L., Y.D., Yanyan Chen, W.Z., and Z.Z. are responsible for the analysis and interpretation of data. J.L., Y.D., Yanyan Chen, J.J., and Yiran Chen are responsible for the writing of the manuscript. H.Z, F.T., Haiyong Wang, Z.Z., and L.T. are responsible for the review and/or revision of the manuscript. Haohao Wang, J.Z., Z.L., Y.L., X.Y., and K.J. are responsible for the administrative, technical, or material support. D.Z., T.Z., Z.Z., Haiyong Wang, and L.T. are responsible for study supervision. All authors read and approved the final manuscript.

## Competing interests

The authors declare no competing interests.
