## [Peer Review File · Nature Communications]

REVIEWER COMMENTS

Reviewer #1, expert in gastric cancer PDX and therapy (Remarks to the Author):

Authors have genetically characterized AFP high gastric cancers using exome sequencing and biomarker assays (IHC). Novelty lies in the patient subgroup chosen and given rarity of this, a small sample size is a good start. Genetic alteration findings such as amplifications in known oncogenes ERBB2/CCNE1 etc background of TP53 mutations adds more to the field. Establishing PDX to show dependence on known oncogenes ERBB2 and CCNE1 in this disease setting is also relevant.

Some comments

Chromosomal instability: Since no cases of EBV and MSI were noted, parallels to the CIN (TCGA) or TP53-loss (ACRG) seems possible. The authors could build upon that further by comparing to similar groups in other GC such as CIN-TCGA/ TP53 Loss in ACRG, rather than comparing only to whole TCGA cohort

MUSIC FDR cutoff seems very liberal at 25%. usually 10-15% is used, which likely is causing a lot more being reported. Fig 1 could use FDR instead of p-value as that perhaps is more relevant. Its not clear if common germline polymorphisms were removed or not from results of exome seq.

Association of amplifications / mutations with outcomes. Several associations are described and not much multivariate analysis is shown to see if these alterations are prognostic after adjusting for known clinical factors such as stage, liver mets etc. Its a little concerning to see several mutations and amplifications show association with prognosis in this small sample set most of which are confounded with TP53 mutation status.

Is Serum AFP level itself associated with poor prognosis as is seeing liver cancers. A analysis related to that may be helpful. Was there any association between seen between mutations/amplifications with serum AFP level itself or serum AFP with prognosis ?

line 134: Noting that ERBB2 and CCNE1 were two essential genes in 17q12 and 19q12..

I would suggest replacing essential genes with well known oncogenes

Not much detail/attention is paid to potential actionability section outside of 2 known oncogenes. for eg BRCA2 is mentioned, but its not clear if the copy gains is a bystander event. It's not clear why BRCA2 copy gain will be a candidate for precision medicine type approach. Also, main text says 14% prevalence of BRCA2 alterations but reduces to 9% in Fig. Multivariate analysis could be helpful to see if ERBB2/CCNE1 were prognostic after adjusting for known clinical prognostic factors.

Patient level metadata and genetic findings seem missing. Not much detail is provided w.r.t. actual mutations and how they are impacting key domains in say some of the top N genes (and how they may differ from GC). freq based comparison appears a bit shallow For eg for TP53, are these in DNA binding domain or elsewhere for SNV that don't lead to deleterious alterations and are these similar to the other non-AFPGC publications?

Reviewer #2, expert in gastric cancer genomics (Remarks to the Author):

I have read the current paper, but I am afraid to describe frankly that the current study might not deserve publication in the current journal. AFPGC is known as GC with abundant expression of serum and tissue AFP, and with severe poor prognosis.

The authors found the higher frequent of the amplification of CCNE(19q12) and ERBB2 (17q12) in AFPGC cases, however, the findings were inevitable. There are bunch of cases of AFPGC without those genomic aberrations. As the authors mentioned that AFPGC indicate the genomic aberrations as a one of the consecutive phenotype, however, it is not the critical and direct cause of AFP producing GC with poor prognosis.

This reviewer thought that the current study has just disclosed that the AFPGC could not be explained by just a genomic alterations. The current journal should demand much definitive findings by adding other phase of study, such as epigenome, transcriptome, proteome and metabolome to disclose the novel findings. This study has been conducted the analysis of relative rare disease pertinently, however, "Nat Commun" should require any revelations, not just a collection of data.

All the bioinformatics method using the specific pipelines were appropriate, however, the current paper does not meet the criteria of the journal, Nat Commun. This paper should be published in the much more specific journal.

Reviewer #3, expert in gastric cancer genomics (Remarks to the Author):

The authors collected a cohort of 105 AFPGC cases among 5261 gastric cancer cases in their institution and performed whole exome sequencing (WES) to examine somatic mutation and copy number alteration information. They detected 19q12 (CCNE1) and 17q12 (ERBB2) amplifications, together with somatic mutation of TP53 as frequent somatic alterations. Then, they used patient derived xenograft (PDX) cell lines established from AFPGC cases to validate the efficacy of molecular targeted therapy, AZD5438 and trastuzumab. Gastric cancer is a heterogeneous disease, and AFPGC is also phenotypically very heterogeneous. Genomic analysis data of AFPGC is a valuable resource to the community, although more detailed data analysis would be required to reveal genomic features of AFPGC. There are several concerns and suggestions.

1. Copy number alteration of CCNE1 and ERBB2 in AFPGC is clearly shown, while these loci are also amplified in non-AFPGC gastric carcinoma, although apparently to lesser extent. TCGA paper (ref. 8) also reported TP53 mutation is enriched in SCNA-high cases, which presumably include AFPGC cases. Taken together, CCNE1 and ERBB2 amplification could be merely a feature of Chromosomally unstable (CIN) subgroup, not necessarily specific to AFPGC. Please show the frequency of these amplifications in CIN type GC in TCGA data and show the effect of copy number status on survival data within CIN subgroup. Ideally, non-AFPGC cases in their cohort could be used as a validation cohort to avoid geographic difference.
2. Survival data in lines 64 to 66 is a little vague, as TCGA-GC can be classified into 4 subtypes. Survival should be compared within CIN subgroup or against other subtypes.
3. Histology of AFPGC is heterogenous, as they also described in Discussion. Please discuss the correlation between genomic alterations and histopathology, e.g. hepatoid adenocarcinoma,

enteroblastic adenocarcinoma, although Hepatoid vs non-hepatoid classification is provided in Fig 1A.

4. To identify genomic features of somatic mutations for AFPGCs, comparison to CIN subgroup would be important, rather than entire TCGA-GC.

5. Mutation signature data in Fig. 1B is interesting. Please provide the figure showing the contribution of each signature, together with histology type. It would be nice to have these information together with Fig. 1A.

6. Regarding the targetable genes in Figure 4B, they used the top 5 genes in AFPGC, which eliminated PIK3CA. Given that a selective inhibitor for mutant PIK3CA is approved for other cancer types, you should include the targetable genes for TCGA-GC.

7. Previous studies on CCNE1 and ERBB2 amplification positive PDX models (doi.org/10.1186/s13045-018-0563-y), where AFP status was not described, demonstrated the efficacy of CDK inhibitor and trastuzumab treatment, respectively. Regarding the survival data in Fig. 4D/E/F, were there any ERBB2 positive cases treated with trastuzumab? If any, how about the treatment response?

Reviewers' comments

Reviewer #1, expert in gastric cancer PDX and therapy (Remarks to the Author):

Authors have genetically characterized AFP high gastric cancers using exome sequencing and biomarker assays (IHC). Novelty lies in the patient subgroup chosen and given rarity of this, a small sample size is a good start. Genetic alteration findings such as amplifications in known oncogenes ERBB2/CCNE1 etc background of TP53 mutations adds more to the field. Establishing PDX to show dependence on known oncogenes ERBB2 and CCNE1 in this disease setting is also relevant.

Some comments

Chromosomal instability: Since no cases of EBV and MSI were noted, parallels to the CIN (TCGA) or TP53-loss (ACRG) seems possible. The authors could build upon that further by comparing to similar groups in other GC such as CIN-TCGA/ TP53 Loss in ACRG, rather than comparing only to whole TCGA cohort

MUSIC FDR cutoff seems very liberal at 25%. usually 10-15% is used, which likely is causing a lot more being reported. Fig 1 could use FDR instead of p-value as that perhaps is more relevant. Its not clear if common germline polymorphisms were removed or not from results of exome seq.

Association of amplifications / mutations with outcomes. Several associations are described and not much multivariate analysis is shown to see if these alterations are prognostic after adjusting for known clinical factors such as stage, liver mets etc. Its a little concerning to see several mutations and amplifications show association with prognosis in this small sample set most of which are confounded with TP53 mutation status.

Is Serum AFP level itself associated with poor prognosis as is seeing liver cancers. A analysis related to that may be helpful. Was there any association between seen between mutations/amplifications with serum AFP level itself or serum AFP with prognosis ?

line 134: Noting that ERBB2 and CCNE1 were two essential genes in 17q12 and 19q12..

I would suggest replacing essential genes with well known oncogenes

Not much detail/attention is paid to potential actionability section outside of 2 known oncogenes. for eg BRCA2 is mentioned, but its not clear if the copy gains is a bystander event. It's not clear why BRCA2 copy gain will be a candidate for precision medicine type approach. Also, main text says 14% prevalence of BRCA2 alterations but reduces to 9% in Fig. Multivariate analysis could be helpful to see if ERBB2/CCNE1 were prognostic after adjusting for known clinical prognostic factors.

Patient level metadata and genetic findings seem missing. Not much detail is provided w.r.t. actual mutations and how they are impacting key domains in say some of the top N genes (and how they may differ from GC). freq based comparison appears a bit shallow For eg for TP53, are these in DNA binding domain or elsewhere for SNV that don't lead to deleterious alterations and are these similar to the other non-AFPGC publications?

Reviewer #2, expert in gastric cancer genomics (Remarks to the Author):

I have read the current paper, but I am afraid to describe frankly that the current study might not deserve publication in the current journal. AFPGC is known as GC with abundant expression of serum and tissue AFP, and with severe poor prognosis.

The authors found the higher frequent of the amplification of CCNE(19q12) and ERBB2 (17q12) in AFPGC cases, however, the findings were inevitable. There are bunch of cases of AFPGC without those genomic aberrations. As the authors mentioned that AFPGC indicate the genomic aberrations as a one of the consecutive phenotype, however, it is not the critical and direct cause of AFP producing GC with poor prognosis.

This reviewer thought that the current study has just disclosed that the AFPGC could not be explained by just a genomic alterations. The current journal should demand much definitive findings by adding other phase of study, such as epigenome, transcriptome, proteome and metabolome to disclose the novel findings. This study has been conducted the analysis of relative rare disease pertinently, however, “Nat Commun” should require any revelations, not just a collection of data.

All the bioinformatics method using the specific pipelines were appropriate, however, the current paper does not meet the criteria of the journal, Nat Commun. This paper should be published in the much more specific journal.

Reviewer #3, expert in gastric cancer genomics (Remarks to the Author):

The authors collected a cohort of 105 AFPGC cases among 5261 gastric cancer cases in their institution and performed whole exome sequencing (WES) to examine somatic mutation and copy number alteration information. They detected 19q12 (CCNE1) and 17q12 (ERBB2) amplifications, together with somatic mutation of TP53 as frequent somatic alterations. Then, they used patient derived xenograft (PDX) cell lines established from AFPGC cases to validate the efficacy of molecular targeted therapy, AZD5438 and trastuzumab. Gastric cancer is a heterogeneous disease, and AFPGC is also phenotypically very heterogeneous. Genomic analysis data of AFPGC is a valuable resource to the community, although more detailed data analysis would be required to reveal genomic features of AFPGC. There are several concerns and suggestions.

1. Copy number alteration of CCNE1 and ERBB2 in AFPGC is clearly shown, while these loci are also amplified in non-AFPGC gastric carcinoma, although apparently to lesser extent. TCGA paper (ref. 8) also reported TP53 mutation is enriched in SCNA-high cases, which presumably include AFPGC cases. Taken together, CCNE1 and ERBB2 amplification could be merely a feature of Chromosomally unstable (CIN) subgroup, not necessarily specific to AFPGC. Please show the frequency of these amplifications in CIN type GC in TCGA data and show the effect of copy number status on survival data within CIN subgroup. Ideally, non-AFPGC cases in their cohort could be used as a validation cohort to avoid geographic difference.
2. Survival data in lines 64 to 66 is a little vague, as TCGA-GC can be classified into 4 subtypes. Survival should be compared within CIN subgroup or against other subtypes.
3. Histology of AFPGC is heterogenous, as they also described in Discussion. Please discuss the correlation between genomic alterations and histopathology, e.g. hepatoid adenocarcinoma, enteroblastic adenocarcinoma, although Hepatoid vs non-hepatoid classification is provided in Fig 1A.
4. To identify genomic features of somatic mutations for AFPGCs, comparison to CIN subgroup

would be important, rather than entire TCGA-GC.

5. Mutation signature data in Fig. 1B is interesting. Please provide the figure showing the contribution of each signature, together with histology type. It would be nice to have these information together with Fig. 1A.

6. Regarding the targetable genes in Figure 4B, they used the top 5 genes in AFP GC, which eliminated PIK3CA. Given that a selective inhibitor for mutant PIK3CA is approved for other cancer types, you should include the targetable genes for TCGA-GC.

7. Previous studies on CCNE1 and ERBB2 amplification positive PDX models (doi.org/10.1186/s13045-018-0563-y), where AFP status was not described, demonstrated the efficacy of CDK inhibitor and trastuzumab treatment, respectively. Regarding the survival data in Fig. 4D/E/F, were there any ERBB2 positive cases treated with trastuzumab? If any, how about the treatment response?

Point-to-point Response

Reviewer #1, expert in gastric cancer PDX and therapy (Remarks to the Author):

Authors have genetically characterized AFP high gastric cancers using exome sequencing and biomarker assays (IHC). Novelty lies in the patient subgroup chosen and given rarity of this, a small sample size is a good start. Genetic alteration findings such as amplifications in known oncogenes ERBB2/CCNE1 etc. background of TP53 mutations adds more to the field. Establishing PDX to show dependence on known oncogenes ERBB2 and CCNE1 in this disease setting is also relevant.

Response: We thank the reviewer for the assessment on the significance and novelty of our study.

Some comments

1. Chromosomal instability: Since no cases of EBV and MSI were noted, parallels to the CIN (TCGA) or TP53-loss (ACRG) seems possible. The authors could build upon that further by comparing to similar groups in other GC such as CIN-TCGA/ TP53 Loss in ACRG, rather than comparing only to whole TCGA cohort

Response: We appreciate the reviewer's suggestions. We deeply agree that comparison with CIN subtype (TCGA) and TP53-loss (ACRG) subtype is interesting and should be taken into consideration in the data analysis. We compared these subtypes from the perspective of significantly mutated genes (SMGs), somatic copy number alterations (SCNAs) and clinical relevance.

1) Comparing with TCGA:

Firstly, we compared the frequency of SMGs between AFPGC and TCGA-GC as well as all four subtypes (**including TCGA-CIN**). As shown in Figure 1B, we found that TP53 mutation in AFPGC was higher than that in TCGA-GC ($P < 0.01$). Furthermore, when comparing to TCGA subtypes, no significant difference in TP53 mutation rate was observed between AFPGC and TCGA-CIN (69% vs 64%, Figure 1C, Supplementary Table S6). Detailed analysis showed that 64.3% TP53 mutations detected in AFPGC occurred in the DNA binding domain, including recurrent missense mutations R273H/C, R272M and R282W (Figure 1D). Additionally, c.994-1G>A (X331_splice), a splice site mutation in the oligomerization domain, was present in 3 cases of AFPGC but absent in TCGA-CIN or TCGA-GC (Figure 1D, Supplementary Fig. S3A). On the contrary, TP53 R175H/G, the most recurrent mutation in TCGA-CIN, was not observed in AFPGC. Besides, the mutation rates of other significantly mutated genes, such as KMT2C, MDC1, FPR1, EPHA1 and SMAD4, were different between AFPGC

and TCGA-CIN (Figure 1C, Supplementary Table S6). We have clarified the above points in the revised manuscript and indicated by highlight (Page 2, Lines 79 to Page 3, Lines 92).

Figure 1B-D | (B-C) Gene mutation rates of AFPGC in comparison with TCGA-GC (B) or four subtypes of TCGA-GC (C).

Orange dots, genes with significantly higher mutation rate in AFPGC; blue dots, genes with significantly lower mutation rate in AFPGC. (D) Distribution of non-synonymous somatic TP53 mutations identified in 58 AFPGC.

Supplementary Fig. S3A | Distribution of non-synonymous somatic mutations in TP53 in AFPGC, TCGA-GC, and TCGA-CIN.

Secondly, the comparison of significant somatic copy number alterations (SCNAs) between AFPGC and TCGA-GC was further performed (Supplementary Fig. S7). A total of 63 significant SCNA regions (28 amplifications and 35 deletions) were identified in AFPGC, of which 34 regions (11 amplifications and 23 deletions) were not significant in TCGA-GC, suggesting a relative specificity for

AFPGC (Supplementary Fig. S8A-B). Moreover, when compared with TCGA-CIN, 43% (12/28) significant amplifications and 74% (26/35) deletions were specific for AFPGC (Supplementary Fig. S8 C-F), such as amplifications in 7q11.21, 6p21.33, 19q13.43 and deletions in 9q34.3, 1p36.32 (Supplementary Fig. S7). These regions contain important cancer-related genes including TRIM28, SEMA3C, NOTCH1, and FAT1. We have clarified the above points in the revised manuscript and indicated by highlight (Page 3, Lines 116-124).

Supplementary Fig. S7 | GISTIC 2.0 significant SCNAs in AFPGC and gastric cancer from TCGA. The comparison of AFPGC and TCGA-GC in amplifications (A) and deletions (B). The comparison of AFPGC and TCGA-GC subtypes in amplifications (C) and deletions (D). Chromosomal locations of peaks of significant focal amplifications (red) and deletions (blue) are plotted by FDR. Annotated regions have an FDR < 0.25, and regions highlighted in red or blue were specific for AFPGC comparing with TCGA-GC or TCGA-CIN. FDR, false discovery rate; AMP, amplification; DEL, deletion.

Supplementary Fig. S8 | Overlap of significant SCNAs between AFPGC and gastric cancer from TCGA. The Venn diagram displays the joint regions in GISTIC 2.0 amplifications (A) and deletions (B) between AFPGC and TCGA-GC, amplifications (C) and deletions (D) between AFPGC and TCGA subtypes, amplifications (E) and deletions (F) between AFPGC and TCGA-CIN subtype.

Thirdly, we compared the frequency of these amplifications between AFPGC and as well as all four subtypes (including TCGA-CIN). The ERBB2 positive rate in AFPGC was significantly higher than that in TCGA-GC and TCGA-CIN (Figure 4D). A similar trend was also observed with regard to CCNE1 amplification (Figure 4D).

Figure 4D | The frequency of ERBB2 and CCNE1 amplification/positive in AFPGC and TCGA-GC. ERBB2-positive status included ERBB2 overexpression (score 3+) by IHC or ERBB2 amplification by FISH as described in previous studies (¹Bang et al, The Lancet, 2010; ²Bartley et al, Journal of Clinical Oncology, 2017). *P* values were from chi-square test or Fisher's exact test. * *P* values were significant.

Moreover, we compared the prognostic value of ERBB2 and CCNE1 between AFPGC and TCGA-CIN. In our previous analysis, we found that ERBB2 or CCNE1 amplification had worse survival outcomes when compared to those with negative status in AFPGC (Figure 4E, G). However, the prognostic value of ERBB2 (Figure 4F) and CCNE1 (Figure 4H) was not observed in TCGA-CIN, which harbors the most frequent ERBB2 and CCNE1 amplification in TCGA subtypes. Interestingly, the co-positive group demonstrated the worst overall survival rate and aggressive biological characteristics (Figure 4I, Supplementary Table S15). Nevertheless, the combined effects of ERBB2 and CCNE1 amplification on prognosis were not observed in TCGA-CIN (Figure 4J). Furthermore, multivariate cox regression analysis demonstrated both ERBB2 and CCNE1 amplification were independent prognostic factors in AFPGC (after adjusting by TP53 mutation status, Figure 4K) but not in TCGA-CIN (Figure 4L).

Figure 4 E-L | (E-F) Associations between ERBB2 status and OS in AFPGC (E) or in TCGA-CIN (F). (G-H) Associations between CCNE1 status and OS in AFPGC (G) or in TCGA-CIN (H). (I-J) Associations between combined status of ERBB2 and CCNE1 and OS in AFPGC (I) or in TCGA-CIN (J). (K-L) Forest plot of multivariable cox proportional hazard regression in AFPGC (K) or in TCGA-CIN (L). POS, positive; AMP, amplification; The status of *ERBB2* was identified as positive when meeting one of the criteria as follows: 1) 3+ expression in IHC; 2) 2+ expression in IHC and amplification in FISH. The amplification of *CCNE1* in FISH was identified as *CCNE1* positive. *ERBB2^N*, *ERBB2* negative; *ERBB2^P*, *ERBB2* positive; *CCNE1^N*, *CCNE1* negative; *CCNE1^P*, *CCNE1* positive; IHC, immunohistochemistry; FISH, fluorescence in situ hybridization; OS, overall survival; HR, hazard ratio; CI, confidence interval.

These comparative analyses revealed that AFPGC has distinct genomic features from gastric cancer of The Cancer Genome Atlas (TCGA) as well as four molecular subtypes (including TCGA-CIN).

2) Comparing with TP53-loss (ACRG):

Firstly, we also compared the frequency of SMGs between AFPGC and TP53-loss subtype (ACRG). As shown in Figure RL1(RL, abbreviation of Response Letter), the mutation rates of SMGs, such as TP53, PRK3CA, KRAS and PTEN, were significantly different between AFPGC and TP53-loss subtype.

Figure RL1 | Comparison of gene mutation frequencies in AFPGC and ACRG-MSS/TP53⁻ subtype. Orange or blue dots represents the genes in AFPGC with significantly higher or lower mutation frequency than those in ACRG-MSS/TP53⁻ subtype.

In addition, the ERBB2-positive rate in AFPGC was significantly higher than that in TCGA-GC and subtypes (Figure RL2). A similar trend was also observed with regard to CCNE1 amplification (Figure RL2).

Figure RL2 | The frequency of ERBB2 and CCNE1 amplifications in AFPGC and gastric cancer from ACRG. ERBB2-positive status included ERBB2 overexpression (score 3+) by IHC or ERBB2 amplification by FISH as described in previous

studies (¹Bang et al, The Lancet, 2010; ²Bartley et al, Journal of Clinical Oncology, 2017). *P* values were from chi-square test or Fisher's exact test. * *P* values were significant.

And thirdly, we explored the prognostic value of ERBB2 and CCNE1 in TP53-loss subtype, whereas no significant association between ERBB2 or CCNE1 amplification and OS was observed in ACRG-MSS/TP53- subtype (Figure RL3).

Figure RL3 | Association between ERBB2 amplification (A) or CCNE1 amplification (B) and OS in ACRG-MSS/TP53- subtype. OS, overall survival; HR, hazard ratio; CI, confidence interval.

In summary, we found that although there are some similarities between AFPGC and TCGA-CIN as well as TP53-loss subtype, there are still distinct differences in genetic characteristics or clinical phenotype. Elucidating the unique properties of AFPGC is a critical step in our efforts for understanding this disease.

2. MUSIC FDR cutoff seems very liberal at 25%. usually 10-15% is used, which likely is causing a lot more being reported. Fig 1 could use FDR instead of p-value as that perhaps is more relevant. It's not clear if common germline polymorphisms were removed or not from results of exome seq.

Response:

1) Thanks for the reviewer's suggestions. We agree with you that it is common in the literature to set the FDR at 0.1-0.15. Given the scarcity of this subtype of gastric cancer, our sample size was relatively limited. In order to avoid omission as much as possible, we raised the threshold of FDR to 0.25. This practice can also be found in other literature, both in cancer-related (³Ellis et al, Nature 2012;

⁴Bismejjer et al, Radiology 2020; ⁵Seo et al, Journal of clinical medicine, 2020) and non-cancer studies (⁶Isganaitis et al, The American journal of clinical nutrition, 2019). For example, Ellis et al identified significantly mutated genes with a FDR < 0.26 using MuSiC package (³Ellis et al, Nature, 2012).

2) Thanks for the reviewer’s suggestion and we have used FDR instead of p-value in Figure 1A.

3) Thanks for the suggestion. In this study, we focused on somatic variants rather than germline polymorphism. To further clarify this point, we have added “Paired adjacent normal tissue DNA was obtained as a control for all tumor samples” in the revised manuscript (**Method part**, Page 10, Line 396-397).

3. Association of amplifications / mutations with outcomes. Several associations are described and not much multivariate analysis is shown to see if these alterations are prognostic after adjusting for known clinical factors such as stage, liver mets etc. Its a little concerning to see several mutations and amplifications show association with prognosis in this small sample set most of which are confounded with TP53 mutation status.

Response: We thank the reviewer for this comment. In the original version, Kaplan-Meier analysis revealed that ERBB2 amplification (P = 0.009; Figure 4E), CCNE1 amplification (P < 0.001; Figure 4G), TP53 mutation (P = 0.045; Supplementary Fig. S11A) and PCLO mutation (P = 0.035; Supplementary Fig. S11F) were prognostic factors in AFPGC. Considering your valuable point that “it is unknown that if these alterations are prognostic after adjusting for known clinical factors”, we further performed multivariate COX analysis, and found that ERBB2 and CCNE1 amplifications were both independent prognostic factors after adjusting for age, TNM stage, serum AFP level, TP53 mutation in 58 cases of AFPGC (ERBB2 amplification: HR = 2.747, P = 0.021; CCNE1 amplification: HR = 2.913, P = 0.010; Figure 4K). Similar results were also observed in 105 cases of AFPGC (ERBB2 amplification: HR = 1.724, P = 0.070; CCNE1 amplification: HR = 2.232, P = 0.006; Figure RL4). Besides, the prognostic value of PCLO mutation was not significant (HR = 1.897, P = 0.121; Figure RL5) after adjusting for age, TNM stage, serum AFP level, and TP53 mutation in multivariate COX analysis.

Figure 4E, G | (E, G) Associations between ERBB2(E) or CCNE1(G) status and OS in AFPGC.

Supplementary Fig. S11 A, F | Survival analysis of significantly mutated genes in AFPGC cohort. Patients were divided into two subgroups according to the mutated status of genes in (A)TP53; (F) PCLO. OS, overall survival.

Figure 4K | Forest plot of multivariable cox proportional hazard regression in 58 cases of AFPGC.

Figure RL4 | Forest plot of multivariable cox proportional hazard regression in 105 cases of AFPGC.

Figure RL5 | Forest plot showing the prognostic value of PCLO mutation status using multivariable cox proportional hazard regression in 58 cases of AFPGC. It was adjusted by age, stage, serum AFP level, and TP53 mutation status.

4. Is Serum AFP level itself associated with poor prognosis as is seeing liver cancers. A analysis related to that may be helpful. Was there any association between seen between mutations/amplifications with serum AFP level itself or serum AFP with prognosis ?

Response:

We thank the reviewer for this comment. We agree that a comparison between prognosis and serum AFP level would be informative. We analyzed the correlation of serum AFP level and overall survival, and 270 ng/mL was chosen as a cutoff value according to ROC curve. The results revealed that patient with high serum AFP level (> 270 ng/mL) had poorer overall survival (Supplementary Fig. S2B-C)

As suggested by the reviewer, we further investigated the correlations between serum AFP level and some significant mutations/amplifications. However, the results revealed that the correlation between mutations/amplifications and serum AFP level was not significant (Table RL1). In addition, the association between serum AFP level and clinicopathological characteristics were shown in Table RL2.

Supplementary Fig. S2B-C | Survival analysis for AFPGC. (B) ROC curve analysis for evaluating the 3-year survival value of serum AFP level. (C) The serum AFP level of AFPGC was associated with overall survival. Low AFP level ($270 \text{ ng/ml} \leq$ serum AFP $\leq 270 \text{ ng/ml}$); High AFP level ($270 \text{ ng/ml} <$ Serum AFP); Non-AFPGC, AFP-negative gastric carcinoma; OS, overall survival; HR, hazard ratio; CI, confidence interval; AUC: area under the ROC curve; ROC: receiver-operating characteristic.

Table RL1. Associations between AFP serum level and genomic alteration frequency in AFPGC.

Gene	Status	Serum AFP level		P
		AFP ≤ 270 ng/ml	AFP > 270 ng/ml	
TP53	No-mut	13	5	0.719
	Mut	27	13	
PCLO	No-mut	33	13	0.486
	Mut	7	5	

CSMD3	No-mut	33	14	0.724*
	Mut	7	4	
KMT2C	No-mut	32	15	>0.999*
	Mut	8	3	
FLG	No-mut	32	16	0.708*
	Mut	8	2	
LRP1B	No-mut	32	16	0.708*
	Mut	8	2	
SYNE1	No-mut	33	15	>0.999*
	Mut	7	3	
COL11A1	No-mut	34	15	>0.999*
	Mut	6	3	
ZFHX4	No-mut	34	15	>0.999*
	Mut	6	3	
MDC1	No-mut	36	15	0.665*
	Mut	4	3	
HCN1	No-mut	36	16	>0.999*
	Mut	4	2	
ERBB2	No-mut	38	15	0.167*
	Mut	2	3	
ERBB4	No-mut	37	16	0.641*
	Mut	3	2	
FPR1	No-mut	36	17	0.67*
	Mut	4	1	
PCDH18	No-mut	37	16	0.641*
	Mut	3	2	
PRDM1	No-mut	40	16	0.093*
	Mut	0	2	
PRKRIR	No-mut	37	16	0.641*
	Mut	3	2	
ERBB2-CNV	No-amp	26	14	0.377*
	Amp	14	4	
CCNE1-CNV	No-amp	29	6	0.758
	Amp	11	12	

*P values were from Fisher's exact test and the others were from chi-square test, and were significant at < 0.05.

Table RL2. Associations between serum AFP level and clinicopathological characteristics in AFPGC

Parameters	Serum AFP level		P
	AFP≤270ng/ml n (%)	AFP>270ng/ml n(%)	
Sex			>0.999
Male	49(75.4%)	30(75.0%)	
Female	16(24.6%)	10(25.0%)	
Age (years)			0.169
≥60	45(69.2%)	33(82.5%)	

<60	20(30.8%)	7(17.5%)	
Tumor location			0.395
Cardia	18(28.1%)	6(16.7%)	
Antrum	34(53.1%)	21(58.3%)	
Body	12(18.8%)	9(25.0%)	
Unspecified	1	4	
TNM stage			0.832
I-II	21(32.3%)	12(30.0%)	
III-IV	44(67.7%)	28(70.0%)	
Histological pattern			0.092
hepatoid	23(35.4%)	23(57.5%)	
Tubular/Papillary	31(47.7%)	12(30.0%)	
Others	11(16.9%)	5(12.5%)	
Lymphovascular invasion			0.816
positive	41(67.2%)	21(70.0%)	
negative	20(32.8%)	9(30.0%)	
NA*	4	10	
Liver metastasis			0.042
Yes	20(30.8%)	22(56.4%)	
No	45(69.2%)	17(43.6%)	
NA**	0	1	

NA, not applicable

*The pathologic report did not indicate the presence of lymphovascular invasion

**No follow-up information was available to confirm the presence of liver metastases

5. line 134: Noting that ERBB2 and CCNE1 were two essential genes in 17q12 and 19q12. I would suggest replacing essential genes with well-known oncogenes

Response: We sincerely thank the reviewer for this advice and amended the statement as suggested in the revised version.

6. Not much detail/attention is paid to potential actionability section outside of 2 known oncogenes. for eg BRCA2 is mentioned, but its not clear if the copy gains is a bystander event. It's not clear why BRCA2 copy gain will be a candidate for precision medicine type approach. Also, main text says 14% prevalence of BRCA2 alterations but reduces to 9% in Fig. Multivariate analysis could be helpful to see if ERBB2/CCNE1 were prognostic after adjusting for known clinical prognostic factors.

Response: We sincerely thank the reviewer for this advice. With regard to potentially actionable gene alterations other than amplification of ERBB2 and CCNE1, we have done additional search in literature,

and representative studies or clinical trials of targeted therapies were summarized in Supplementary Table S11.

Supplementary Table S11. Target therapies of gene alterations and relevant clinical trials in cancer treatment

Gene	Drug	Clinical trials or preclinical studies (cancer type)	Phase
AKT2	Afuresertib	NCT04374630 (ovarian cancer)	Phase 2
AURKA	BPR1K871	PMID: 27863392 (acute myeloid leukemia)	Preclinical
AXL	Bemcentinib	NCT03824080 (acute myeloid leukemia)	Phase 2
BCL6	FX1	PMID: 27482887 (lymphoma)	Preclinical
BRCA2	Niraparib	NCT04475939 (non small cell lung cancer)	Phase 3
		NCT04235101 (solid tumor)	Phase 1
CCND1	Abemaciclib	NCT04584853 (breast cancer)	Phase 3
		NCT04238819 (relapsed solid tumor)	Phase 1
CCND3	Abemaciclib	NCT04584853 (breast cancer)	Phase 3
		NCT04238819 (relapsed solid tumor)	Phase 1
CCNE1	Dinaciclib	NCT01580228 (chronic lymphocytic leukemia)	Phase 3
		NCT01434316 (solid tumors)	Phase 1
CDK6	Abemaciclib	NCT04584853 (breast cancer)	Phase 3
		NCT04238819 (relapsed solid tumor)	Phase 1
EGFR	Gefitinib, Erlotinib, Afatinib	Non small cell lung cancer	FDA approved
ERBB2	Trastuzumab	Breast cancer, gastric cancer	FDA approved
ERBB3	Sapitinib	NCT01579578 (metastatic gastric cancer)	Phase 2
ERBB4	Afatinib	Non small cell lung cancer	FDA approved
FGFR2	Futibatinib	NCT04024436 (metastatic breast cancer)	Phase 2
FLT1	Pazopanib HCl	Advanced renal cell carcinoma and soft tissue sarcoma	FDA approved
FLT3	Gilteritinib	NCT02752035/NCT04027309 (acute myeloid leukemia)	Phase 3
IGF1R	Brigatinib	NCT04111705 (metastatic non small cell lung cancer)	Phase 2
MCL1	AZD5991	NCT03218683 (hematologic malignancy)	Phase 1
MYC	MYCi361	PMID: 31679823 (solid tumor)	Preclinical
PAK1	IPA-3	PMID: 32240651 (prostate cancer)	Preclinical
PIK3CA	CH5132799	NCT01222546 (advanced solid tumors)	Phase 1

We apologize for the confusion on actionability of gene BRCA2 .We agree with the reviewer that there is currently no evidence for the actionability of BRCA2 amplification or copy number gain, nor for other alterations such as deletion of ERBB2, AXL or IGF1R. Therefore, to avoid confusion, we removed these alterations in the revised Figure 4A and the alteration rates in AFPGC and TCGA-GC were correspondingly changed.

Figure 4A | Genomic alterations of potentially targetable genes in AFPGC and TCGA-GC.

In Figure 4A, we intended to visualize all alterations of actionable genes in AFPGC. As for the issue with BRCA2, in the original version, 14% was its alteration rate and 9% was the mutation rate in AFPGC cohort. In the revised version, we removed BRCA2 amplification and the alteration rates in AFPGC and TCGA-GC were correspondingly changed.

As the reviewer suggested, we have performed multivariate COX analysis and found that ERBB2 and CCNE1 amplifications were both independent prognostic factors after adjusting for age, TNM stage, serum AFP level, TP53 mutation in AFPGC (Figure 4K, Figure RL4; see response to Question 1 and 3).

Figure 4K | Forest plot of multivariable cox proportional hazard regression in 58 cases of AFPGC.

Figure RL4 | Forest plot of multivariable cox proportional hazard regression in 105 cases of AFPGC.

7. Patient level metadata and genetic findings seem missing. Not much detail is provided w.r.t. actual mutations and how they are impacting key domains in say some of the top N genes (and how they may differ from GC). freq based comparison appears a bit shallow For eg for TP53, are these in DNA binding domain or elsewhere for SNV that don't lead to deleterious alterations and are these similar to the other non-AFPGC publications?

Response:

We are grateful for the reviewer's suggestion. In this revision, we added detailed patient-level clinicopathological data (Supplementary Table S1) and provided the alteration information in the source data. According to your suggestion, we depicted the mutation distribution in 4 frequently mutated genes, TP53, KMT2C, FPR1 and EPHA1, in AFPGC, in comparison with TCGA-GC and TCGA-CIN (Supplementary Fig. S3). For example, in TP53, 64.3% mutations occurred in the DNA binding domain. Interestingly, splice site mutation c.994-1G>A (X331_splice) in the oligomerization domain was present in 3 cases of AFPGC but absent in TCGA-CIN (Supplementary Fig. S3A). This splice site mutation was predicted as deleterious by MutationTaster. On the other hand, TP53 R175H/G, the most recurrent mutation in TCGA-CIN, was not observed in AFPGC.

Supplementary Fig. S3 | Distribution of non-synonymous somatic mutations in frequently mutated genes in AFPGC, TCGA-GC, and TCGA-CIN. (A) TP53; (B) KMT2C; (C) FPR1; (D) EPHA1.

We sincerely appreciate the reviewer's kind comments and the suggestions have tremendously helped us improve the depth of our analysis.

Reviewer #2, expert in gastric cancer genomics (Remarks to the Author):

I have read the current paper, but I am afraid to describe frankly that the current study might not deserve publication in the current journal. AFPGC is known as GC with abundant expression of serum and tissue AFP, and with severe poor prognosis.

The authors found the higher frequent of the amplification of CCNE1(19q12) and ERBB2 (17q12) in AFPGC cases, however, the findings were inevitable. There are bunch of cases of AFPGC without those genomic aberrations. As the authors mentioned that AFPGC indicate the genomic aberrations as a one of the consecutive phenotype, however, it is not the critical and direct cause of AFP producing GC with poor prognosis.

Response: We thank the reviewer for this comment and agree with the reviewer's suggestion. Gastric cancer is a highly heterogeneous disease. Although AFPGC is a special subtype of gastric cancer, there still exists heterogeneity to a certain extent. In this study, we mainly focus on elucidating the major genomic features and facilitating the development of specialized therapies for this disease. More in-depth research on heterogeneity will be the future direction. Our study revealed higher frequency of the amplification of CCNE1(19q12) and ERBB2 (17q12) observed in AFPGC, compared with TCGA-GC. In this revised version (Figure 4E-L), we compared the prognostic value of ERBB2 and CCNE1 between AFPGC and TCGA-CIN (which most frequently harbors ERBB2 and CCNE1 amplification among four TCGA subtypes). Multivariate COX regression analysis demonstrated that ERBB2 and CCNE1 amplification were independent prognostic factors in AFPGC but not in TCGA-CIN, implying a unique role of these alterations in the development and progression of AFPGC.

Figure 4 E-L | (E-F) Associations between ERBB2 status and OS in AFPGC (E) or in TCGA-CIN (F). (G-H) Associations between CCNE1 status and OS in AFPGC (G) or in TCGA-CIN (H). (I-J) Associations between combined status of ERBB2 and CCNE1 and OS in AFPGC (I) or in TCGA-CIN (J). (K-L) Forest plot of multivariable cox proportional hazard regression in AFPGC (K) or in TCGA-CIN (L). POS, positive; AMP, amplification; The status of *ERBB2* was identified as positive when meeting one of the criteria as follows: 1) 3+ expression in IHC; 2) 2+ expression in IHC and amplification in FISH. The amplification of *CCNE1* in FISH was identified as *CCNE1* positive. *ERBB2*^N, *ERBB2* negative; *ERBB2*^P, *ERBB2* positive; *CCNE1*^N, *CCNE1* negative; *CCNE1*^P, *CCNE1* positive; IHC, immunohistochemistry; FISH, fluorescence in situ hybridization; OS, overall survival; HR, hazard ratio; CI, confidence interval.

This reviewer thought that the current study has just disclosed that the AFPGC could not be explained by just a genomic alterations. The current journal should demand much definitive findings by adding other phase of study, such as epigenome, transcriptome, proteome and metabolome to disclose the novel findings. This study has been conducted the analysis of relative rare disease pertinently, however, “Nat Commun” should require any revelations, not just a collection of data.

Response: We deeply agree that multi-omics study including epigenome, transcriptome, proteome and metabolome, could largely enhance our understanding of this disease. However, as this was a retrospective study, the quality of samples could not meet the criteria of multi-omics sequencing. Multi-omics study through prospective research design will be future direction. In this revised version, we have added these statements in our limitation part page 8, line 338-339. Moreover, we performed in-depth analyses according to the reviewer’s suggestion. Comparative genomic analyses between AFPGC and TCGA-GC cohort (including four molecular subtypes) were carried out from the perspective of significantly mutated genes (Figure 1B-D) and somatic copy number alterations (Supplementary Fig. S7, Supplementary Fig. S8). These additional analyses confirmed that AFPGC has distinct genomic features from gastric cancer of The Cancer Genome Atlas (TCGA) or each of its four molecular subtypes.

Figure 1B-D | (B-C) Gene mutation rates of AFPGC in comparison with TCGA-GC (B) or four subtypes of TCGA-GC (C).

Orange dots, genes with significantly higher mutation rate in AFPGC; blue dots, genes with significantly lower mutation rate in

APFGC. (D) Distribution of non-synonymous somatic TP53 mutations identified in 58 AFPGC.

Supplementary Fig. S7 | GISTIC 2.0 significant SCNAs in AFPGC and gastric cancer from TCGA. The comparison of AFPGC and TCGA-GC in amplifications (A) and deletions (B). The comparison of AFPGC and TCGA-GC subtypes in amplifications (C) and deletions (D). Chromosomal locations of peaks of significant focal amplifications (red) and deletions (blue) are plotted by FDR. Annotated regions have an FDR < 0.25, and regions highlighted in red or blue were specific for AFPGC comparing with TCGA-GC or TCGA-CIN. FDR, false discovery rate; AMP, amplification; DEL, deletion.

Supplementary Fig. S8 | Overlap of significant SCNAs between AFPGC and gastric cancer from TCGA. The Venn diagram displays the joint regions in GISTIC 2.0 amplifications (A) and deletions (B) between AFPGC and TCGA-GC, amplifications (C) and deletions (D) between AFPGC and TCGA subtypes, amplifications (E) and deletions (F) between AFPGC and TCGA-CIN subtype.

All the bioinformatics method using the specific pipelines were appropriate, however, the current paper does not meet the criteria of the journal, Nat Commun. This paper should be published in the much more specific journal.

Response: We thank the reviewer for the positive comments on our bioinformatics method. AFPGC is a rare subtype of gastric cancer associated with high liver metastasis rate (nearly 40% in this study) and poor prognosis. Till date, very little is known regarding the molecular characteristics of AFPGC, which seriously restricted the understanding and management of this disease. To the best of our knowledge, this is the largest genomic study upon AFPGC. Our

study revealed the distinct genomic characteristics of AFPGC, which is a critical contribution to the research of this disease. Besides, the PDX models applied in this study could be an ideal platform to identify potential molecular targets and optimize therapeutic approaches for AFPGC. Considering the novelty and the potential clinical value of this study, we believe that this manuscript fits into the scope of Nature Communications. Once again, we thank the reviewer for the efforts in reviewing our manuscript. Please find our revisions in the re-submitted files.

Reviewer #3, expert in gastric cancer genomics (Remarks to the Author):

The authors collected a cohort of 105 AFPGC cases among 5261 gastric cancer cases in their institution and performed whole exome sequencing (WES) to examine somatic mutation and copy number alteration information. They detected 19q12 (CCNE1) and 17q12 (ERBB2) amplifications, together with somatic mutation of TP53 as frequent somatic alterations. Then, they used patient derived xenograft (PDX) cell lines established from AFPGC cases to validate the efficacy of molecular targeted therapy, AZD5438 and trastuzumab. Gastric cancer is a heterogeneous disease, and AFPGC is also phenotypically very heterogeneous. Genomic analysis data of AFPGC is a valuable resource to the community, although more detailed data analysis would be required to reveal genomic features of AFPGC. There are several concerns and suggestions.

1. Copy number alteration of CCNE1 and ERBB2 in AFPGC is clearly shown, while these loci are also amplified in non-AFPGC gastric carcinoma, although apparently to lesser extent. TCGA paper (ref. 8) also reported TP53 mutation is enriched in SCNA-high cases, which presumably include AFPGC cases. Taken together, CCNE1 and ERBB2 amplification could be merely a feature of Chromosomally unstable (CIN) subgroup, not necessarily specific to AFPGC. Please show the frequency of these amplifications in CIN type GC in TCGA data and show the effect of copy number status on survival data within CIN subgroup. Ideally, non-AFPGC cases in their cohort could be used as a validation cohort to avoid geographic difference.

Response: We appreciate the reviewer's suggestions. We agree that comparison with CIN subtype (TCGA) is interesting and should be taken into consideration in our analysis. Firstly, we compared the frequency of these amplifications between AFPGC and as well as all four subtypes (including TCGA-CIN). The ERBB2 positive rate in AFPGC was significantly higher than that in TCGA-GC and TCGA-CIN (Figure 4D). A similar trend was also observed with regard to CCNE1 amplification (Figure 4D). Secondly, we compared the prognostic value of ERBB2 and CCNE1 amplification between AFPGC and TCGA-GC as well as all four subtypes. In our previous analysis, we found that patients with ERBB2 or CCNE1 amplification had worse survival outcomes when compared to those with negative status in AFPGC (Figure 4E, G). However, the prognostic value of ERBB2 (Figure 4F) and CCNE1 (Figure 4H) was not observed in TCGA-CIN. Interestingly, the co-positive group demonstrated the worst overall survival rate and aggressive biological characteristics in AFPGC (Figure 4I, Supplementary Table S15). Nevertheless, the combined effects of ERBB2 and CCNE1 amplification on prognosis were not observed in TCGA-CIN (Figure 4J). Furthermore, multivariate cox regression analysis demonstrated both ERBB2

and *CCNE1* amplification were independent prognostic factors in AFPGC (after adjusting by *TP53* mutation status, Figure 4K) but not in TCGA-CIN (Figure 4L).

Figure 4D | The frequency of *ERBB2* and *CCNE1* amplification/positive in AFPGC and TCGA-GC. *ERBB2*-positive status included *ERBB2* overexpression (score 3+) by IHC or *ERBB2* amplification by FISH as described in previous studies (¹Bang et al, The Lancet, 2010; ²Bartley et al, Journal of Clinical Oncology, 2017). *P* values were from chi-square test or Fisher's exact test. * *P* values were significant.

Figure 4 E-L | (E-F) Associations between *ERBB2* status and OS in AFPGC (E) or in TCGA-CIN (F). (G-H) Associations between *CCNE1* status and OS in AFPGC (G) or in TCGA-CIN (H). (I-J) Associations between combined status of *ERBB2* and *CCNE1* and OS in AFPGC (I) or in TCGA-CIN (J). (K-L) Forest plot of multivariable cox proportional hazard regression in AFPGC (K) or in TCGA-CIN (L). POS, positive; AMP, amplification; The status of *ERBB2* was identified as positive when meeting one of the criteria as follows: 1) 3+ expression in IHC; 2) 2+ expression in IHC and amplification in FISH. The amplification of *CCNE1* in FISH was identified as *CCNE1* positive. *ERBB2*^N, *ERBB2* negative; *ERBB2*^P, *ERBB2*

positive; CCNE1^N, *CCNE1* negative; CCNE1^P, *CCNE1* positive; IHC, immunohistochemistry; FISH, fluorescence in situ hybridization; OS, overall survival; HR, hazard ratio; CI, confidence interval.

Additionally, the comparison of significant somatic copy number alterations (SCNAs) between AFPGC and TCGA-GC was also performed (Supplementary Fig. S7). A total of 63 significant SCNA regions (28 amplifications and 35 deletions) were identified in AFPGC, of which 34 regions (11 amplifications and 23 deletions) were not significant in TCGA-GC, suggesting a relative specificity for AFPGC (Supplementary Fig. S8A-B). Moreover, when compared with TCGA-CIN, 43% (12/28) significant amplifications and 74% (26/35) deletions were specific for AFPGC (Supplementary Fig. S8C-F), such as amplifications in 7q11.21, 6p21.33, 19q13.43 and deletions in 9q34.3, 1p36.32 (Supplementary Fig. S7). These regions contain important cancer-related genes including TRIM28, SEMA3C, NOTCH1, and FAT1. We have clarified the above points in the revised manuscript and indicated by highlight (Page 3, Lines 116-124).

In summary, we found that although there are some similarities between AFPGC and TCGA-CIN, there are still distinct differences in genetic characteristics or clinical phenotype. Elucidating the unique properties of AFPGC is a critical step in our efforts for understanding this disease.

Supplementary Fig. S7 | GISTIC 2.0 significant SCNAs in AFPGC and gastric cancer from TCGA. The comparison of AFPGC and TCGA-GC in amplifications (A) and deletions (B). The comparison of AFPGC and TCGA-GC subtypes in amplifications (C) and deletions (D). Chromosomal locations of peaks of significant focal amplifications (red) and deletions (blue) are plotted by FDR. Annotated regions have an FDR < 0.25, and regions highlighted in red or blue were specific for AFPGC comparing with TCGA-GC or TCGA-CIN. FDR, false discovery rate; AMP, amplification; DEL, deletion.

Supplementary Fig. S8 | Overlap of significant SCNAs between AFPGC and gastric cancer from TCGA. The Venn diagram displays the joint regions in GISTIC 2.0 amplifications (A) and deletions (B) between AFPGC and TCGA-GC, amplifications (C) and deletions (D) between AFPGC and TCGA subtypes, amplifications (E) and deletions (F) between AFPGC and TCGA-CIN subtype.

As for using non-AFPGC cases to avoid geographic difference, we very much agree with your suggestion, and we are very interested in the comparison results. At present, our research group is preparing to match and sequencing non-AFPGC cases and looking forward to future studies.

2. Survival data in lines 64 to 66 is a little vague, as TCGA-GC can be classified into 4 subtypes. Survival should be compared within CIN subgroup or against other subtypes.

Response: We thank the reviewer for this comment. Our intention was to compare the prognosis of AFPGC and non-AFPGC in our institution. In our previous description, we did not label the source of the control group clearly which might lead to misunderstanding. We apologize for our negligence, and we clarified this issue in our revised manuscript. Please find the corresponding revision on page 2, line 64-65.

3. Histology of AFPGC is heterogenous, as they also described in Discussion. Please discuss the correlation between genomic alterations and histopathology, e.g. hepatoid adenocarcinoma, enteroblastic adenocarcinoma, although Hepatoid vs non-hepatoid classification is provided in Fig 1A.

Response: We thank the reviewer for this comment. In the literature, AFPGC can be classified histologically into hepatoid adenocarcinoma and other differentiated adenocarcinomas such as tubular/papillary adenocarcinoma, enteroblastic adenocarcinoma and yolk-sac tumor-like carcinoma. In clinical practice, some histological subtypes, such as enteroblastic adenocarcinoma and yolk-sac tumor-like carcinoma are not included in the WHO classification of digestive system tumors (5th edition, 2019) as well as lacking confirmed definition. Therefore, in this study, histological subtypes of tumor samples were re-evaluated according to the WHO histological classification system. Finally, the histopathology of AFPGC in this study was classified as follows: hepatoid adenocarcinoma, tubular/papillary adenocarcinoma and other types (Supplementary Table S1). We further compared the frequency of significantly mutated gene among different pathological subtypes, but no significant result (Table RL3) was observed. We altered Figure 1A and the corresponding figure legend. Moreover, the distribution of significant mutation signatures across different pathological subtypes were also analyzed (Figure 1F, Supplementary Fig. S5). We also discussed these points in the limitation on page 8, line 336-337. Although this study represented the largest genomic analyses of AFPGC samples so far, our analyses were still limited by sample size. That is, the results, such as significantly mutated genes, chromosomal alterations, and the correlation between genetic alterations and clinicopathological variables (e.g histopathology, TNM stage) should still be fully validated in a larger multi-centered cohort,

Table RL3. Associations between histopathology and genomic alteration frequency in AFPGC.

Gene	Status	Histopathology			P
		Hepatoid	Tubular/Papillary	Others	
TP53	No-mut	9	7	2	0.864*
	Mut	18	15	7	
PCLO	No-mut	19	18	9	0.175*
	Mut	8	4	0	
CSMD3	No-mut	20	20	7	0.324*
	Mut	7	2	2	
KMT2C	No-mut	22	19	6	0.439*
	Mut	5	3	3	
FLG	No-mut	24	16	8	0.339*
	Mut	3	6	1	
LRP1B	No-mut	22	17	9	0.380*
	Mut	5	5	0	

SYNE1	No-mut	24	17	7	0.565*
	Mut	3	5	2	
COL11A1	No-mut	23	18	8	>0.999*
	Mut	4	4	1	
ZFHX4	No-mut	24	16	9	0.152*
	Mut	3	6	0	
MDC1	No-mut	22	21	8	0.298*
	Mut	5	1	1	
HCN1	No-mut	24	20	8	>0.999*
	Mut	3	2	1	
ERBB2	No-mut	24	21	8	0.693*
	Mut	3	1	1	
ERBB4	No-mut	25	19	9	0.567*
	Mut	2	3	0	
FPR1	No-mut	26	20	7	0.222*
	Mut	1	2	2	
PCDH18	No-mut	26	19	8	0.365*
	Mut	1	3	1	
PRDM1	No-mut	25	19	9	0.567*
	Mut	2	3	0	
PRKRIR	No-mut	25	19	9	0.567*
	Mut	2	3	0	
ERBB2-CNV	No-amp	18	17	5	0.474*
	Amp	9	5	4	
CCNE1-CNV	No-amp	20	17	4	0.177*
	Amp	7	5	5	

*P values were from Fisher's exact test and the others were from chi-square test, and were significant at < 0.05 .

4. To identify genomic features of somatic mutations for AFPGCs, comparison to CIN subgroup would be important, rather than entire TCGA-GC.

Response: Thanks for the reviewer's suggestion. In the revised version, we compared the frequency of significantly mutated genes between AFPGC and TCGA-GC as well as all four subtypes (**including TCGA-CIN**). As shown in Figure 1B, we found that TP53 mutation in AFPGC was higher than that in TCGA-GC ($P < 0.01$). Furthermore, when comparing to TCGA subtypes, no significant difference in TP53 mutation rate was observed between AFPGC and TCGA-CIN (69% vs 64%, Figure 1C, Supplementary Table S6). Detailed analysis showed that 64.3% TP53 mutations detected in AFPGC occurred in the DNA binding domain, including recurrent missense mutations R273H/C, R272M and R282W (Figure 1D). Additionally, c.994-1G>A (X331_splice), a splice site mutation in the oligomerization domain, was present in 3 cases of AFPGC but absent in TCGA-CIN or TCGA-GC

(Figure 1D, Supplementary Fig. S3). On the contrary, TP53 R175H/G, the most recurrent mutation in TCGA-CIN, was not observed in AFPGC. Besides, the mutation rates of other significantly mutated genes, such as KMT2C, MDC1, FPR1, EPHA1 and SMAD4, were different between AFPGC and TCGA-CIN (Figure 1C, Supplementary Table S6). We have clarified the above points in the revised manuscript and indicated by highlight (Page 2, Line 79 to page 3, line 92).

Figure 1B-D | (B-C) Gene mutation rates of AFPGC in comparison with TCGA-GC (B) or four subtypes of TCGA-GC (C). Orange dots, genes with significantly higher mutation rate in AFPGC; blue dots, genes with significantly lower mutation rate in AFPGC. (D) Distribution of non-synonymous somatic TP53 mutations identified in 58 AFPGC.

Supplementary Fig. S3A | Distribution of non-synonymous somatic mutations in TP53 in AFPGC, TCGA-GC, and TCGA-CIN.

5. Mutation signature data in Fig. 1B is interesting. Please provide the Fig. Showing the contribution of each signature, together with histology type. It would be nice to have these information together with Fig. 1A.

Response: We appreciate the reviewer’s constructive suggestion. We characterized the contributions of mutation signatures (Signature 1, 17, and 29) in AFPGC across various clinicopathological characteristics (Figure 1F). Statistical analyses revealed that there was no significant relevance between the mutation signatures and the clinical pathological characteristics, including histological types, age, gender and TNM stage (All $P > 0.05$, Supplementary Fig. S5). We have clarified these points in the revised manuscript and indicated by highlight (Page 3, Line 101-103).

Figure 1F | The distribution of mutation signatures in AFPGC across various clinicopathological characteristics. AFPGC, alpha-fetoprotein producing gastric carcinoma.

Supplementary Fig. S5 | Association between signature contributions and clinicopathological characteristics. (A) Histology type; (B) Gender; (C) Age; (D) TNM stage. $P < 0.05$ was considered significant

6. Regarding the targetable genes in Figure 4B, they used the top 5 genes in AFP GC, which eliminated PIK3CA. Given that a selective inhibitor for mutant PIK3CA is approved for other cancer types, you should include the targetable genes for TCGA-GC.

Response:

Thanks for the reviewer's suggestion. We agree with the reviewer that PIK3CA is an important targetable gene in common gastric cancer, and it is also frequently mutated in TCGA-GC (16.5%). However, in Figure 4B, we intended to display 5 most frequently altered targetable genes in AFP GC and their frequency of alteration in TCGA-GC, whereas PIK3CA (7% amplification and no mutation in AFP GC) was not among them.

Figure 4B | Overall frequency of selected targetable genes in AFPGC and TCGA-GC.

We did additional work in literature review and made modification in the display of targetable genes. Alterations with no evidence of actionability, such as amplification of BRCA2, deletion of ERBB2, AXL or IGF1R were removed from Figure 4A in the revision and the alteration rates in AFPGC and TCGA-GC were correspondingly changed.

Figure 4A | Genomic alterations of potentially targetable genes in AFPGC and TCGA-GC.

For all the displayed genes in Figure 4A, we added detailed information about representative studies or clinical trials of targeted therapies, summarized in Supplementary Table S11.

Supplementary Table S11. Target therapies of gene alterations and relevant clinical trials in cancer treatment

Gene	Drug	Clinical trials or preclinical studies (cancer type)	Phase
AKT2	Afuresertib	NCT04374630 (ovarian cancer)	Phase 2
AURKA	BPR1K871	PMID: 27863392 (acute myeloid leukemia)	Preclinical
AXL	Bemcentinib	NCT03824080 (acute myeloid leukemia)	Phase 2
BCL6	FX1	PMID: 27482887 (lymphoma)	Preclinical
BRCA2	Niraparib	NCT04475939 (non small cell lung cancer)	Phase 3

			NCT04235101 (solid tumor)	Phase 1
CCND1	Abemaciclib		NCT04584853 (breast cancer)	Phase 3
			NCT04238819 (relapsed solid tumor)	Phase 1
CCND3	Abemaciclib		NCT04584853 (breast cancer)	Phase 3
			NCT04238819 (relapsed solid tumor)	Phase 1
CCNE1	Dinaciclib		NCT01580228 (chronic lymphocytic leukemia)	Phase 3
			NCT01434316 (solid tumors)	Phase 1
CDK6	Abemaciclib		NCT04584853 (breast cancer)	Phase 3
			NCT04238819 (relapsed solid tumor)	Phase 1
EGFR	Gefitinib, Erlotinib, Afatinib		Non small cell lung cancer	FDA approved
ERBB2	Trastuzumab		Breast cancer, gastric cancer	FDA approved
ERBB3	Sapitinib		NCT01579578 (metastatic gastric cancer)	Phase 2
ERBB4	Afatinib		Non small cell lung cancer	FDA approved
FGFR2	Futibatinib		NCT04024436 (metastatic breast cancer)	Phase 2
FLT1	Pazopanib HCl		Advanced renal cell carcinoma and soft tissue sarcoma	FDA approved
FLT3	Gilteritinib		NCT02752035/NCT04027309 (acute myeloid leukemia)	Phase 3
IGF1R	Brigatinib		NCT04111705 (metastatic non small cell lung cancer)	Phase 2
MCL1	AZD5991		NCT03218683 (hematologic malignancy)	Phase 1
MYC	MYC361		PMID: 31679823 (solid tumor)	Preclinical
PAK1	IPA-3		PMID: 32240651 (prostate cancer)	Preclinical
PIK3CA	CH5132799		NCT01222546 (advanced solid tumors)	Phase 1

7. Previous studies on CCNE1 and ERBB2 amplification positive PDX models (doi.org/10.1186/s13045-018-0563-y), where AFP status was not described, demonstrated the efficacy of CDK inhibitor and trastuzumab treatment, respectively. Regarding the survival data in Fig. 4D/E/F, were there any ERBB2 positive cases treated with trastuzumab? If any, how about the treatment response?

Response: We thank the reviewer for this comment. In our AFPGC cohort, there are 19 metastatic AFPGC patients with ERBB2 positive (Herceptin is approved for metastatic gastric cancer by FDA), of which 6 patients received Herceptin treatment. Then we performed survival analysis comparing trastuzumab therapy in ERBB2-positive AFPGC and ERBB2-positive non- AFPGC (7 patients from our non-AFPGC control group). The result suggests that there is no significant difference in overall survival between ERBB2-positive AFPGC patients and ERBB2-positive non-AFPGC treated with trastuzumab ($P = 0.763$, Figure RL6A). We further compared the overall survival between ERBB2-positive AFPGC patients with and those without Herceptin treatment ($P = 0.257$, Figure RL6B). In this study, a very limited number of patients were enrolled and the proportion of patients receiving Herceptin treatment is relatively low, due to the delayed approval of trastuzumab by the National Medical Products Administration (NMPA) of China and relatively low affordability of Chinese patients. Therefore, it's

difficult to draw definite conclusions on survival benefits from trastuzumab therapy. Although no positive results were observed in this study, we still believe that the question raised by the reviewer is very constructive and worthy of further exploration in larger cohorts. In this revision, we have added these points in our limitation part as follows: “limited by the delayed approval of trastuzumab by the National Medical Products Administration (NMPA) of China and relatively low affordability of Chinese patients, the proportion of patients who received trastuzumab treatment in this study was relatively low, leading to insufficient cases to evaluate the efficacy of ERBB2-targeted therapy” (Page 8, Line 338-343).

Figure RL6 | Survival analysis of ERBB2-targeted therapy in ERBB2-positive AFPGC, comparing to ERBB2-positive non-AFPGC patients with ERBB2-targeted therapy (A) or comparing to ERBB2-positive AFPGC patients without ERBB2-targeted therapy (B). Tra, trastuzumab (ERBB2-targeted inhibitor); OS, overall survival; HR, hazard ratio; CI, confidence interval.

Reference

- 1 Bang, Y.-J. *et al.* Trastuzumab in combination with chemotherapy versus chemotherapy alone for treatment of HER2-positive advanced gastric or gastro-oesophageal junction cancer (ToGA): a phase 3, open-label, randomised controlled trial. *The Lancet* **376**, 687-697, doi:10.1016/s0140-6736(10)61121-x (2010).
- 2 Bartley, A. N. *et al.* HER2 Testing and Clinical Decision Making in Gastroesophageal Adenocarcinoma: Guideline From the College of American Pathologists, American Society for Clinical Pathology, and the American Society of Clinical Oncology. *Journal of Clinical Oncology* **35**, 446-464, doi:10.1200/jco.2016.69.4836 (2017).

- 3 Ellis, M. J. *et al.* Whole-genome analysis informs breast cancer response to aromatase inhibition. *Nature* **486**, 353-360, doi:10.1038/nature11143 (2012).
- 4 Bismeyer, T. *et al.* Radiogenomic Analysis of Breast Cancer by Linking MRI Phenotypes with Tumor Gene Expression. *Radiology* **296**, 277-287, doi:10.1148/radiol.2020191453 (2020).
- 5 Seo, E. S. *et al.* Metastatic Burden Defines Clinically and Biologically Distinct Subgroups of Stage 4 High-Risk Neuroblastoma. *Journal of clinical medicine* **9**, doi:10.3390/jcm9092730 (2020).
- 6 Isganaitis, E. *et al.* Maternal obesity and the human milk metabolome: associations with infant body composition and postnatal weight gain. *The American journal of clinical nutrition* **110**, 111-120, doi:10.1093/ajcn/nqy334 (2019).

REVIEWER COMMENTS

Reviewer #1 (Remarks to the Author):

The manuscript has been modified to address the questions/suggestions and appears to have improved substantially on the topic of AFP+ GC being a different subgroup of GC and also likely different that TCGA CIN or TP53-/MSS ACRG. Other interesting results are now noted such as higher AFP within this group tend to show poorer prognosis in a manner similar to what is know for HCC. Additionally, the multivariate regression results help showcase the prognostic association findings in a much more clear manner than before.

Some stylistic / presentation suggestions are listed to (potentially) help better orient the readers.

1. Better framing of comparison with TCGA CIN or MSS/TP53- could help the readers. For eg see line 80 onwards in the current submission where the authors have compared to TCGA-CIN subtype without providing much context on why that was done, with the key point is the relatively higher number of copy number alterations in background of mutant TP53 background compared to the other known subtypes (fig S7C)
2. The paper may benefit from having a separate section comparative comparing APFGC with subgroups within TCGA / ACRG rather than meshing the results with AFP-GC along with TCGA comparison in the same section (or explicitly provide a rationale early on why this comparison is being done).
3. Its not clear why results from ACRG dataset are provided in the response but omitted form the manuscript / supplemental information. The clinical covariate comparison in a manner similar to Fig4 K/L could make this comprehensive , even if reported in supplemental sections.
4. 19q12 17q12 amplification and association with survival seems redundant as results from ERBB2 and CCNE1 have already been included and no. other gene from that locus was mentioned. perhaps moving this to supplemental section along with results of other loci may help.

Reviewer #3 (Remarks to the Author):

In this revised manuscript, the authors added the comparison analysis with TCGA subtypes according to the request in my previous review. However, the results they provided here were not quite convincing enough to demonstrate that ERBB2 and CCNE1 amplification is specific in AFPGC, as those events are also frequent in TCGA-CIN subtype. In genomic profiling, AFPGC has several genomic alterations common with TCGA-CIN, such as TP53 mutation and ERBB2/CCNE1 amplification, but somehow AFPGC transdifferentiates to display the hepatic phenotype, although the mechanism is still unknown.

1. Copy number alteration of CCNE1 and ERBB2.

In Fig 4D, ERBB2 positivity is not significantly different between AFPGC-WES and TCGA-CIN, although it is barely higher in AFPGC-total ($p=0.041$). Same with CCNE1 amplification.

In Fig 4E and G, ERBB2 and CCNE1 status was analyzed by IHC and/or FISH, while, in TCGA data set

(Fig. 4F and H), genomic amplification is apparently used (details are not shown). The authors should clarify how they scored the ERBB2 and CCNE1 status. To compare their data to TCGA data, the authors should score the amplification status of ERBB2 and CCNE1 genes in their AFPGC samples. Or you could also use the gene expression data based on TCGA RNA-seq results as the expression status of ERBB2 and CCNE1.

In Fig. S7A, AFPGC data appear to be simply presented in the order of exon capture probe locations, therefore, the chromosomal length is not accurate. So, it should be drawn according to actual genomic positions for easier comparison. I would say copy number alteration pattern are rather similar between AFP-GC and TCGA-CIN, although that of AFPGC is a little simpler, which does not contradict that AFPGC is a subgroup of TCGA-CIN.

2. Survival data

The authors can still classify your control samples into EBV, MSI, gastric and intestinal subtypes.

3. Histological subtypes

No further comment

4. Genomic mutation features

Most of the differentially mutated genes reported by the authors are rather infrequent, namely in the long tail. Distribution of TP53 mutations in AFPGC appears not so much different from that in TCGA-GC data.

Regarding the mutation positions, they pointed out that TP53 R175 mutation is not found in AFPGC, but did not provide any implication of their observation.

Taken together, there are no significant difference in somatic mutations specific to AFPGC.

5. Mutation signature

No further comment

6. Targetable gene

No further comment

7. Trastuzumab efficacy in PDX and human

No further comment. Interesting to see the response in a larger cohort, as the survival appears a little better in Trastuzumab treatment group in Figure RL.6.

Reviewers' comments

Reviewer #1 (Remarks to the Author):

The manuscript has been modified to address the questions/suggestions and appears to have improved substantially on the topic of AFP+ GC being a different subgroup of GC and also likely different than TCGA CIN or TP53-/MSS ACRG. Other interesting results are now noted such as higher AFP within this group tend to show poorer prognosis in a manner similar to what is known for HCC. Additionally, the multivariate regression results help showcase the prognostic association findings in a much more clear manner than before.

Some stylistic / presentation suggestions are listed to (potentially) help better orient the readers.

1. Better framing of comparison with TCGA CIN or MSS/TP53- could help the readers. For eg see line 80 onwards in the current submission where the authors have compared to TCGA-CIN subtype without providing much context on why that was done, with the key point is the relatively higher number of copy number alterations in background of mutant TP53 background compared to the other known subtypes (fig S7C)
2. The paper may benefit from having a separate section comparative comparing APFGC with subgroups within TCGA / ACRG rather than meshing the results with AFP-GC along with TCGA comparison in the same section (or explicitly provide a rationale early on why this comparison is being done).
3. It's not clear why results from ACRG dataset are provided in the response but omitted from the manuscript / supplemental information. The clinical covariate comparison in a manner similar to Fig4 K/L could make this comprehensive, even if reported in supplemental sections.
4. 19q12 17q12 amplification and association with survival seems redundant as results from ERBB2 and CCNE1 have already been included and no other gene from that locus was mentioned. Perhaps moving this to supplemental section along with results of other loci may help.

Reviewer #3 (Remarks to the Author):

In this revised manuscript, the authors added the comparison analysis with TCGA subtypes according to the request in my previous review. However, the results they provided here were not quite convincing enough to demonstrate that ERBB2 and CCNE1 amplification is specific in APFGC, as those events are also frequent in TCGA-CIN subtype. In genomic profiling, APFGC has several genomic alterations common with TCGA-CIN, such as TP53 mutation and ERBB2/CCNE1 amplification, but somehow APFGC transdifferentiates to display the hepatic phenotype, although the mechanism is still unknown.

1. Copy number alteration of CCNE1 and ERBB2.

In Fig 4D, ERBB2 positivity is not significantly different between AFPGC-WES and TCGA-CIN, although it is barely higher in AFPGC-total ($p=0.041$). Same with CCNE1 amplification.

In Fig 4E and G, ERBB2 and CCNE1 status was analyzed by IHC and/or FISH, while, in TCGA data set (Fig. 4F and H), genomic amplification is apparently used (details are not shown). The authors should clarify how they scored the ERBB2 and CCNE1 status. To compare their data to TCGA data, the authors should score the amplification status of ERBB2 and CCNE1 genes in their AFPGC samples. Or you could also use the gene expression data based on TCGA RNA-seq results as the expression status of ERBB2 and CCNE1.

In Fig. S7A, AFPGC data appear to be simply presented in the order of exon capture probe locations, therefore, the chromosomal length is not accurate. So, it should be drawn according to actual genomic positions for easier comparison. I would say copy number alteration pattern are rather similar between AFP-GC and TCGA-CIN, although that of AFPGC is a little simpler, which does not contradict that AFPGC is a subgroup of TCGA-CIN.

2. Survival data

The authors can still classify your control samples into EBV, MSI, gastric and intestinal subtypes.

3. Histological subtypes

No further comment

4. Genomic mutation features

Most of the differentially mutated genes reported by the authors are rather infrequent, namely in the long tail. Distribution of TP53 mutations in AFPGC appears not so much different from that in TCGA-GC data.

Regarding the mutation positions, they pointed out that TP53 R175 mutation is not found in AFPGC, but did not provide any implication of their observation.

Taken together, there are no significant difference in somatic mutations specific to AFPGC.

5. Mutation signature

No further comment

6. Targetable gene

No further comment

7. Trastuzumab efficacy in PDX and human

No further comment. Interesting to see the response in a larger cohort, as the survival appears a little better in Trastuzumab treatment group in Figure RL.6.

Point-to-point Response

Reviewer #1 (Remarks to the Author):

The manuscript has been modified to address the questions/suggestions and appears to have improved substantially on the topic of AFP+ GC being a different subgroup of GC and also likely different that TCGA CIN or TP53-/MSS ACRG. Other interesting results are now noted such as higher AFP within this group tend to show poorer prognosis in a manner similar to what is know for HCC. Additionally, the multivariate regression results help showcase the prognostic association findings in a much more clear manner than before.

Response: We sincerely thank the reviewer for the comment and highly appreciate the reviewer's valuable suggestions to improve our study.

Some stylistic / presentation suggestions are listed to (potentially) help better orient the readers.

1. Better framing of comparison with TCGA CIN or MSS/TP53- could help the readers. For eg see line 80 onwards in the current submission where the authors have compared to TCGA-CIN subtype without providing much context on why that was done, with the key point is the relatively higher number of copy number alterations in background of mutant TP53 background compared to the other known subtypes (fig S7C)

Response: Thanks for the reviewer's valuable advice. As suggested, we have revised the context in these parts as follows.

(Page 2, Lines 82-84) “Furthermore, when comparing to TCGA subtypes, TP53 mutation rate of AFPGC was significantly higher than all subtypes other than TCGA-CIN (Figure 1C, Supplementary Table S6).”

(Page 3, Lines 119-122) “Considering that AFPGC harbored relatively higher number of TP53 mutation rate and SCNAs, we subsequently compared the SCNAs of AFPGC with TCGA-CIN (a known subtype with recurrent SCNAs in TCGA-GC), finding that 57% (24/42) significant amplifications and 74% (37/50) deletions were only existed in AFPGC (Supplementary Fig. S8C-F) ...”

(Page 4, Lines 174-176) “Although ERBB2 and CCNE1 amplification were also enriched in TCGA-CIN, the combined effects of ERBB2 and CCNE1 on prognosis were not observed (Figure 4J).”

Thanks again for the reviewer's detailed suggestion.

2. The paper may benefit from having a separate section comparative comparing APFGC with subgroups within TCGA / ACRG rather than meshing the results with AFP-GC along with TCGA comparison in the same section (or explicitly provide a rationale early on why this comparison is being done).

Response: Thanks for the reviewer's suggestions. As suggested, we have provided the rationale on why this comparison is being done between AFPGC and TCGC-GC subtypes in the revised version (Page 2, Lines 82-84; Page 3, Lines 119-122; Page 4, Lines 174-176). In the discussion section, we have extensively revised a separate paragraph to discuss AFPGC and TCGA-CIN.(Page 7, Lines 274-292) “In this study, we found that AFPGC presented with recurrent copy number alterations. Considering chromosomal instability is a hallmark for TCGA-CIN, we, therefore, compared the significant SCNA profiles between AFPGC and TCGA-CIN...”

3. Its not clear why results from ACRG dataset are provided in the response but omitted form the manuscript / supplemental information. The clinical covariate comparison in a manner similar to Fig4 K/L could make this comprehensive, even if reported in supplemental sections.

Response: Thanks for the reviewer’s opinion. According to the reviewer’s suggestion, we have supplemented the results from ACRG datasets in the supplemental information and added this part of discussion in the revised version (Page 7, Lines 306-310)

As for “the clinical covariate comparison in a manner similar to Fig4 K/L”, we further performed the multivariate COX analysis for ERBB2 and CCNE1 amplification in the ACRG cohort. It was demonstrated that patients with ERBB2 amplification harbored more favorable OS in ACRG MSS/TP53-subtype, whereas the survival value of CCNE1 amplification was not significant (Supplementary Fig. S13E). As a contrast, patients with *ERBB2* or *CCNE1* amplification both had worse survival outcomes when compared to those with no amplification in AFPGC, which was different from the results in the ACRG cohort. It was consistent with the results of Kaplan-Meier analysis in the previous version. We also added these results into supplemental sections (Supplementary Fig. S13). We are grateful for the reviewer’s valuable suggestion.

E

Supplementary Fig. S13E. Forest plot of multivariable cox proportional hazard regression in ACRG-MSS/TP53- subtype.

K

Figure 4K. Forest plot of multivariable cox proportional hazard regression in AFPGC.

4. 19q12 17q12 amplification and association with survival seems redundant as results from ERBB2 and CCNE1 have already been included and no. other gene from that locus was mentioned. perhaps moving this to supplemental section along with results of other loci may help.

Response: Thanks for the suggestions. According to the reviewer’s valuable comment, we have moved these results to Supplementary Figure S6, and have made the corresponding modification in the revised manuscript. (Page 3, Line 113)

Reviewer #3 (Remarks to the Author):

In this revised manuscript, the authors added the comparison analysis with TCGA subtypes according to the request in my previous review. However, the results they provided here were not quite convincing enough to demonstrate that ERBB2 and CCNE1 amplification is specific in AFPGC, as those events are also frequent in TCGA-CIN subtype. In genomic profiling, AFPGC has several genomic alterations common with TCGA-CIN, such as TP53 mutation and ERBB2/CCNE1 amplification, but somehow AFPGC transdifferentiates to display the hepatic phenotype, although the mechanism is still unknown.

Response: We thank the reviewer very much for the time in reviewing our revised manuscript. Alpha-fetoprotein producing gastric carcinoma (AFPGC) is a rare and special subtype of gastric cancer (GC) associated with a high liver metastasis rate (nearly 50%) and extremely poor prognosis. The unique clinical characteristics of AFPGC were described by this study and several other studies (¹ Hirasaki, S. et al, Intern Med, 2004; ² Gong, W. et al, Neoplasma, 2018). Our study aims to take a first step towards evaluating the molecular characteristics of AFPGC and provides references for understanding this disease and better clinical management.

In this study, we found that ERBB2 and CCNE1 were the top two most frequent SCN alterations in the AFPGC cohort. Further comparison with the TCGA-CIN (a known subtype with recurrent copy number alterations, the most abundant subtype of ERBB2 and CCNE1 in TCGA-GC), we also observed that ERBB2 and CCNE1 tended to be amplified more frequently in AFPGC-total. From the perspective of the difference in frequency, we agree with the reviewer that it is not quite convincing enough to demonstrate the specific of ERBB2 /CCNE1 amplification in AFPGC. Actually, we found that AFPGC patients with ERBB2 and/or CCNE1 amplification showed a higher liver metastasis rate and worse prognosis. As a contrast, such phenomenon was not found in the TCGA-CIN subtype. Thus, we believe that the ERBB2/CCNE1 amplification is an important molecular event in AFPGC. Subsequent AFPGC PDX experiment also validated the translational significance of the above findings. These pre-clinical data are encouraging and may yield new insights for better management of this disease.

As the reviewer concerned, we also found that AFPGC is chromosomally instable and shared some molecular events with TCGA-CIN, including some SCNAs and mutations. Chromosomal instability (CIN) is a common feature of gastrointestinal adenocarcinomas (GIAC) (³⁻⁵Cancer Genome Atlas Research Network, Nature, 2012, 2014, 2017) and nearly 70 % of GIAC can be classified into CIN subtype (⁶Liu, Y. et al, Cancer Cell, 2018). Recent studies further classified CIN subtype of GC as well as other GIAC into subclasses with different genomic alterations and clinical significance (⁶Liu, Y. et al, Cancer Cell, 2018, ⁷Turajlic, S. et al, Nature Review Genetics, 2019), suggesting that CIN subtype of GC may still be a heterogeneous group. A recent study also demonstrated that CIN is a driver of metastatic progression (⁸Bakhoun, S. F. et al, Nature, 2018). This might partially contribute to the aggressive phenotype of AFPGC. In addition, Liu, Y et al. observed that metastasis from CIN-high tumor cells tended to involved multiple organs. Similarly, we also found that CIN subtype of GC displayed a multi-organ metastatic pattern by analyzing an MSK cohort reported by Janjigian Y and his colleagues (⁹ Janjigian, Y. Y. et al, Cancer Discovery, 2018). However, AFPGC showed a dominant tendency of liver metastasis (Supplementary Fig. S12), suggesting the different metastatic patterns between AFPGC and CIN subtype of GC. In all, the role of chromosomal instability in AFPGC is interesting and worth

further exploration in larger cohorts by multi-omics in the future.

In the revised version, we have added a separate section for comparing AFPGC with TCGA-CIN and made a corresponding modification (Page 7, Lines 274-292).

We thank the reviewer for the comment and highly appreciate the reviewer's valuable suggestions to dramatically improve our study.

Supplementary Fig. S12. Comparison of metastatic pattern between MSKGC-CIN subtype (A) and AFPGC (B). The data of CIN subtype of GC was derived from MSKCC, Cancer Discovery 2017)(download from cBioportal database). liver, liver metastasis only; peritoneum, peritoneum metastasis only; ovary, ovary metastasis only; pleura, pleura metastasis only; brain, brain metastasis only; bone, bone metastasis only; multiple, multi-organ metastasis.

1. Copy number alteration of CCNE1 and ERBB2.

In Fig 4D, ERBB2 positivity is not significantly different between AFPGC-WES and TCGA-CIN, although it is barely higher in AFPGC-total (p=0.041). Same with CCNE1 amplification.

Response: Thanks for the reviewer's comment. In this study, we found that ERBB2 and CCNE1 were the top two most frequent SCNA alterations in AFPGC through whole-exome sequencing. Although this study represented the largest genomic analyses of AFPGC samples so far, our analyses were still limited by sample size. To better illustrate our results on ERBB2 and CCNE1, we validated the amplification status in our clinical sample by FISH (N=105). Further comparison showed that AFPGC tended to have a higher frequency of amplification of ERBB2 and CCNE1 than TCGA-CIN (the most abundant subtype of ERBB2 and CCNE1 in TCGA-GC). More importantly, except for the fact that AFPGC enriched with ERBB2 and CCNE1 amplification, we also observed the phenomenon that AFPGC patients with ERBB2 and/or CCNE1 amplification were closely related to the poor prognosis and liver metastasis of patients. However, such phenomenon was not found in the TCGA-CIN cohort. These findings suggested that ERBB2 or CCNE1 amplification might play an important role in the tumor genesis and progression of AFPGC. The corresponding results and discussions have been illustrated in Page 4, Lines 154-178.

In Fig 4E and G, ERBB2 and CCNE1 status was analyzed by IHC and/or FISH, while, in TCGA data set (Fig. 4F and H), genomic amplification is apparently used (details are not shown). The authors should clarify how they scored the ERBB2 and CCNE1 status. To compare their data to TCGA data, the authors should score the amplification status of ERBB2 and CCNE1 genes in their AFPGC samples. Or you could also use the gene expression data based on TCGA RNA-seq results as the expression status of ERBB2 and CCNE1.

Response: We thank the reviewer for this comment. We apologize for the confusion caused by the direct comparison between our previous FISH/IHC data and TCGA data.

In the last version, we evaluated ERBB2 status according to the National Comprehensive Cancer Network (NCCN) guidelines, which recommend assessment of ERBB2 overexpression using immunohistochemistry (IHC) and ERBB2 amplification using fluorescence in situ hybridization (FISH) or in situ hybridization (ISH). The positive status of ERBB2 was defined as ERBB2 IHC (3+) or ERBB2 IHC (2+) with amplification by FISH. Positive (IHC 3+) or negative (IHC 0 or 1+) of ERBB2 status do not require further FISH testing (Type: evidence based; Quality of evidence: high; Strength of recommendation: strong). When ERBB2 status is equivocal (IHC 2+), FISH testing should be further performed to test the status of ERBB2 amplification.

In this revised version, to make a better comparison, we further performed FISH testing on ERBB2 (IHC 3+) samples (22 cases) and found that all these samples showed ERBB2 amplification. Previous studies also reported that the correlation between ERBB2 IHC 3+ and ERBB2 amplification in FISH was highly concordant [94%-100% (¹⁰Bang, Y. J. et al, Lancet, 2010; ¹¹Bartley, A. N. et al, Journal of Clinical Oncology, 2017)]. As for CCNE1, we tested the status of CCNE1 amplification using FISH testing but not IHC in our previous version.

In the revised version, we have made the corresponding modifications in the legend of Figure 4D and Supplementary Table S13-15. Thanks again for the reviewer's valuable suggestions to improve our work.

In Fig. S7A, AFPGC data appear to be simply presented in the order of exon capture probe locations, therefore, the chromosomal length is not accurate. So, it should be drawn according to actual genomic positions for easier comparison. I would say copy number alteration pattern are rather similar between AFP-GC and TCGA-CIN, although that of AFPGC is a little simpler, which does not contradict that AFPGC is a subgroup of TCGA-CIN.

Response: Thanks for the reviewer's detailed suggestion. To revise, we used the same marker_file in GISTIC 2.0 to compare the SCNAs between AFPGC and TCGA-GC, and redrew the Supplementary Figure S7A and Figure S8, as suggested by the reviewer. The results might be slightly different with the previous version which used marker_file with exon capture probe locations for AFPGC. We agree with the reviewer that the SCNA pattern is partly similar between AFPGC and TCGA-CIN, but there were still several regions with amplifications or deletions that differ between AFPGC and TCGA-CIN. We have modified the description for the comparison of AFPGC with TCGA-GC and TCGA-CIN in the revised version (Page 3, Lines 115-122). The relationship between AFPGC and CIN is worth further exploration in the future.

Supplementary Fig. S7. GISTIC 2.0 significant SCNAs in AFPGC and gastric cancer from TCGA. The comparison of AFPGC and TCGA-GC in amplifications (A) and deletions (B). The comparison of AFPGC and TCGA-GC subtypes in amplifications (C) and deletions (D). Chromosomal locations of peaks of significant focal amplifications (red) and deletions (blue) are plotted by FDR. Annotated regions have an FDR < 0.25, and regions highlighted in red or blue were specific for AFPGC comparing with TCGA-GC or TCGA-CIN. FDR, false discovery rate; AMP, amplification; DEL, deletion.

Supplementary Fig. S8. Overlap of significant SCNAs between AFPGC and gastric cancer from TCGA. The Venn diagram displays the joint regions in GISTIC 2.0 amplifications (A) and deletions (B) between AFPGC and TCGA-GC, amplifications (C) and deletions (D) between AFPGC and TCGA subtypes, amplifications (E) and deletions (F) between AFPGC and TCGA-CIN subtype.

2. Survival data

The authors can still classify your control samples into EBV, MSI, gastric and intestinal subtypes.

Response: We thank the reviewer for this comment. Since 2017, NCCN guidelines for GC have gradually recommended MSI and EBV as potential biomarkers for personalized treatment in GC. Currently, in clinical practice, MSI status is assessed by IHC staining to measure expression levels of proteins involved in DNA mismatch repair (MLH1, MSH2, MSH6, PMS2), and EBV status is detected by EBER in situ hybridization¹² (Setia, N. et al. Modern pathology, 2016). The classification of gastric and intestinal subtypes is mainly based on the expression of mucins protein, including gastric marker mucins (MUC5AC, MUC6, MUC1) and intestinal marker molecules (MUC2 and CD10). However, both in NCCN and Chinese Society of Clinical Oncology (CSCO) guidelines, these mucin types (gastric and intestinal) are not currently recommended for routine clinical practice in our country.

Recently, Massachusetts General Hospital proposed five molecular subgroups of gastric adenocarcinoma using IHC and *in situ* hybridization, including Epstein–Barr virus (EBV) positivity, microsatellite instability (MSI), aberrant E-cadherin expression, aberrant p53 expression, and normal p53 expression. Aberrant p53 expression cluster was subdivided into four subgroups based on mucin typing (gastric and intestinal), showing differences in nodal metastasis and lymphovascular invasion

between subcategories. Regrettably, mucin type has not been included in routine typing of GC in clinical practice. Therefore, most patients in our control group have not been tested for mucins protein.

According to your suggestion, we further analyzed 44 patients diagnosed after 2017 in the control group. Among them, 2 cases were EBV positive and 2 cases were MSI-H.

In addition, considering Lauren classification is routinely recommended in our clinical practice, we tried to further analyze the survival difference between AFPGC and different Lauren subtypes of our control group. It was shown that AFPGC harbored a more unsatisfactory prognosis (Figure RL1), which needed to be validated in larger multi-centered cohort.

Figure RL1. Comparison of overall survival between AFPGC and different Lauren subtypes of stage-matched non-AFPGC.

3. Histological subtypes

No further comment

Response: Thanks for the reviewer's approval.

4. Genomic mutation features

Most of the differentially mutated genes reported by the authors are rather infrequent, namely in the long tail.

Distribution of TP53 mutations in AFPGC appears not so much different from that in TCGA-GC data.

Regarding the mutation positions, they pointed out that TP53 R175 mutation is not found in AFPGC, but did not provide any implication of their observation.

Taken together, there are no significant difference in somatic mutations specific to AFPGC.

Response: Thanks for the reviewer's comment. The differentially mutated genes shown in Figure 1 were indeed mostly infrequent. Firstly, this analysis only included significantly mutated genes (SMGs) identified in AFPGC or TCGA, most of which in AFPGC were infrequent (<10%). Secondly, considering that the low incidence of AFPGC has limited the sample size of our cohort, the difference of SMGs between AFPGC and TCGA-GC may not reach statistical significance. This part has been supplemented in our limitations (Page 8, Lines 346-350). A recent study illustrated that some genes in the long tail might still be potential oncogenic drivers (¹³Armenia, J. et al, Nature Genetics, 2019).

As a complement to SMG-related analysis, we provide a comparison of the frequently mutated genes (>10%) between AFPGC and TCGA-GC/TCGA-CIN. Here we screened out several genes such as TP53 and KMT2C which were significantly higher than those in TCGA-GC (Supplementary Table

S17) or TCGA-CIN (Supplementary Table S18). Therefore, there are still some notable differences in frequently mutated genes between AFPGC and TCGA-GC as well as TCGA-CIN, some of which have been reported to play an important role in tumorigenesis and progression (Supplementary Table S17-18).

Regarding the mutation positions of TP53, cases in AFPGC harbored X331_splice mutation, which was absent in TCGA-GC but present in other cancer types including lung, colon and liver cancer¹⁴(Bailey, M. H. et al, Cell, 2018). This mutation was predicted as “disease causing” by MutationTaster¹⁵ (Schwarz et al, Nature methods, 2010) and may induce loss of capacity for oligomerization, leading to partial or complete loss of transactivation potential of p53¹⁶(Sabapathy, K. & La et al, Nature reviews Clinical oncology, 2018) . Notably, as the most frequent hotspot in TCGA-GC (or TCGA-CIN), TP53 R175 mutation is absent in AFPGC. This mutation was reported to reduce the DNA binding capacity of p53 (loss-of-function) and was oncogenic in multiple cancers including GC¹⁷.(Bullock, A. N et al, Oncogene, 2000). Our comparison revealed that there were some important differences in TP53 mutation positions between AFPGC and TCGA-GC.

We have added related discussion in this revised manuscript (Page 6, Lines 235-242)

Figure RL2. Mutation rates of frequently mutated genes in AFPGC comparing to TCGA-GC (A) or TCGA-CIN (B).

5. Mutation signature

No further comment

Response: Thanks for the reviewer’s approval.

6. Targetable gene

No further comment

Response: Thanks for the reviewer’s approval.

7. Trastuzumab efficacy in PDX and human

No further comment. Interesting to see the response in a larger cohort, as the survival appears a little better in Trastuzumab treatment group in Figure RL.6.

Response: Thanks for the reviewer’s approval.

- with endoscopic mucosal resection and additional surgery. *Intern Med* **43**, 926-930, doi:10.2169/internalmedicine.43.926 (2004).
- 2 Gong, W. *et al.* Clinical characteristics and treatments of patients with alpha-fetoprotein producing gastric carcinoma. *Neoplasma* **65**, 326-330, doi:10.4149/neo_2018_170207N84 (2018).
- 3 Cancer Genome Atlas Research, N. Comprehensive molecular characterization of gastric adenocarcinoma. *Nature* **513**, 202-209, doi:10.1038/nature13480 (2014).
- 4 Cancer Genome Atlas, N. Comprehensive molecular characterization of human colon and rectal cancer. *Nature* **487**, 330-337, doi:10.1038/nature11252 (2012).
- 5 Cancer Genome Atlas Research, N. *et al.* Integrated genomic characterization of oesophageal carcinoma. *Nature* **541**, 169-175, doi:10.1038/nature20805 (2017).
- 6 Liu, Y. *et al.* Comparative Molecular Analysis of Gastrointestinal Adenocarcinomas. *Cancer Cell* **33**, 721-735 e728, doi:10.1016/j.ccell.2018.03.010 (2018).
- 7 Turajlic, S., Sottoriva, A., Graham, T. & Swanton, C. Resolving genetic heterogeneity in cancer. *Nat Rev Genet* **20**, 404-416, doi:10.1038/s41576-019-0114-6 (2019).
- 8 Bakhoun, S. F. *et al.* Chromosomal instability drives metastasis through a cytosolic DNA response. *Nature* **553**, 467-472, doi:10.1038/nature25432 (2018).
- 9 Janjigian, Y. Y. *et al.* Genetic Predictors of Response to Systemic Therapy in Esophagogastric Cancer. *Cancer Discov* **8**, 49-58, doi:10.1158/2159-8290.CD-17-0787 (2018).
- 10 Bang, Y. J. *et al.* Trastuzumab in combination with chemotherapy versus chemotherapy alone for treatment of HER2-positive advanced gastric or gastro-oesophageal junction cancer (ToGA): a phase 3, open-label, randomised controlled trial. *Lancet* **376**, 687-697, doi:10.1016/S0140-6736(10)61121-X (2010).
- 11 Bartley, A. N. *et al.* HER2 Testing and Clinical Decision Making in Gastroesophageal Adenocarcinoma: Guideline From the College of American Pathologists, American Society for Clinical Pathology, and the American Society of Clinical Oncology. *J Clin Oncol* **35**, 446-464, doi:10.1200/JCO.2016.69.4836 (2017).
- 12 Setia, N. *et al.* A protein and mRNA expression-based classification of gastric cancer. *Modern pathology : an official journal of the United States and Canadian Academy of Pathology, Inc* **29**, 772-784, doi:10.1038/modpathol.2016.55 (2016).
- 13 Armenia, J. *et al.* Publisher Correction: The long tail of oncogenic drivers in prostate cancer. *Nat Genet* **51**, 1194, doi:10.1038/s41588-019-0451-6 (2019).
- 14 Bailey, M. H. *et al.* Comprehensive Characterization of Cancer Driver Genes and Mutations. *Cell* **174**, 1034-1035, doi:10.1016/j.cell.2018.07.034 (2018).
- 15 Schwarz, J. M., Rodelsperger, C., Schuelke, M. & Seelow, D. MutationTaster evaluates disease-causing potential of sequence alterations. *Nat Methods* **7**, 575-576, doi:10.1038/nmeth0810-575 (2010).
- 16 Sabapathy, K. & Lane, D. P. Therapeutic targeting of p53: all mutants are equal, but some mutants are more equal than others. *Nature reviews. Clinical oncology* **15**, 13-30, doi:10.1038/nrclinonc.2017.151 (2018).
- 17 Bullock, A. N., Henckel, J. & Fersht, A. R. Quantitative analysis of residual folding and DNA binding in mutant p53 core domain: definition of mutant states for rescue in cancer therapy. *Oncogene* **19**, 1245-1256, doi:10.1038/sj.onc.1203434 (2000).

REVIEWERS' COMMENTS

Reviewer #3 (Remarks to the Author):

The concerns I raised on the previous version have been appropriately answered with adding experimental data and data analysis.